# Diet-induced loss of adipose hexokinase 2 correlates with hyperglycemia

**Mitsugu Shimobayashi[1,2]\*, Amandine Thomas[1], Sunil Shetty[1], Irina C Frei[1], Bettina K Wölnerhanssen[3,4], Diana Weissenberger[1], Anke Vandekeere[5,6], Mélanie Planque[5,6], Nikolaus Dietz[1], Danilo Ritz[1], Anne Christin Meyer-Gerspach[3,4], Timm Maier[1], Nissim Hay[7], Ralph Peterli[8], Sarah-Maria Fendt[5,6], Nicolas Rohner[9,10], Michael N Hall[1]\***

[1]Biozentrum, University of Basel, Basel, Switzerland; [2]Department of Chronic Diseases and Metabolism, Laboratory of Clinical and Experimental Endocrinology, KU Leuven, Leuven, Belgium; [3]University of Basel, Basel, Switzerland; [4]St. Clara Research Ltd, St. Claraspital, Basel, Switzerland; [5]Laboratory of Cellular Metabolism and Metabolic Regulation, VIB-KU Leuven Center for Cancer Biology, Leuven, Belgium; [6]Department of Oncology, Laboratory of Cellular Metabolism and Metabolic Regulation, KU Leuven and Leuven Cancer Institute, Leuven, Belgium; [7]Department of Biochemistry and Molecular Genetics, College of Medicine, University of Illinois at Chicago, Chicago, United States; [8]Clarunis, Department of Visceral Surgery, University Centre for Gastrointestinal and Liver Diseases, Basel, Switzerland; [9]Stowers Institute for Medical Research, Kansas City, United States; [10]Department of Cell Biology and Physiology at the University of Kansas School of Medicine, Kansas City, United States

**\*For correspondence:**
mitsugu.shimobayashi@
kuleuven.be (MS);
m.hall@unibas.ch (MNH)

**Abstract** Chronically high blood glucose (hyperglycemia) leads to diabetes and fatty liver disease. Obesity is a major risk factor for hyperglycemia, but the underlying mechanism is unknown. Here, we show that a high-fat diet (HFD) in mice causes early loss of expression of the glycolytic enzyme Hexokinase 2 (HK2) specifically in adipose tissue. Adipose-specific knockout of *Hk2* reduced glucose disposal and lipogenesis and enhanced fatty acid release in adipose tissue. In a non-cell-autonomous manner, *Hk2* knockout also promoted glucose production in liver. Furthermore, we observed reduced hexokinase activity in adipose tissue of obese and diabetic patients, and identified a loss-of-function mutation in the *hk2* gene of naturally hyperglycemic Mexican cavefish. Mechanistically, HFD in mice led to loss of HK2 by inhibiting translation of *Hk2* mRNA. Our findings identify adipose HK2 as a critical mediator of local and systemic glucose homeostasis, and suggest that obesity-induced loss of adipose HK2 is an evolutionarily conserved mechanism for the development of selective insulin resistance and thereby hyperglycemia.

## Editor's evaluation

This study reveals that expression of the glycolytic enzyme hexokinase 2 (HK2) in adipocytes is decreased in obesity and is associated with glucose intolerance and insulin resistance. The authors then show that adipose selective depletion of HK2 in mice causes systemic glucose intolerance, suggesting that the decreased HK2 may contribute to metabolic dysfunction in obese humans. These studies point to a potentially important new pathway that contributes to the regulation of metabolic health.

## Introduction

Vertebrates mediate glucose homeostasis by regulating glucose production and disposal in specific tissues (*Roden and Shulman, 2019*; *Wasserman, 2009*). High blood glucose stimulates pancreatic beta cells to secrete the hormone insulin which in turn promotes glucose disposal in skeletal muscle and adipose tissue and inhibits glucose production in liver. Although its contribution to glucose clearance is minor (*Jackson et al., 1986*; *Kowalski and Bruce, 2014*), adipose tissue plays a particularly important role in systemic glucose homeostasis (*Abel et al., 2001*; *Shepherd et al., 1993*). Adipose-specific knockout of insulin signaling components, such as the insulin receptor, mTORC2, and AKT, results in local and systemic insulin insensitivity (*Beg et al., 2017*; *Cybulski et al., 2009*; *Frei et al., 2022*; *Jiang et al., 2003*; *Kumar et al., 2010*; *Sakaguchi et al., 2017*; *Shearin et al., 2016*; *Tang et al., 2016*). However, we and others have reported that diet-induced obesity in mice causes adipose dysfunction, systemic insulin insensitivity, and hyperglycemia despite normal insulin signaling in white adipose tissue (WAT; *Figure 1—figure supplement 1A*; *Shimobayashi et al., 2018*; *Tan et al., 2015*). How does obesity cause systemic insulin insensitivity and hyperglycemia? In other words, how does diet induce hyperglycemia?

Here, we show that a high-fat diet induces early loss of the glycolytic enzyme HK2 specifically in adipose tissue. Loss of adipose HK2 leads to reduced glucose disposal by adipose tissue and increased glucose production in liver, ultimately causing glucose intolerance. Loss of adipose HK2 also decreased lipogenesis and increased fatty acid release in WAT. This and related findings in Mexican cavefish and adipose tissue of obese patients suggest that diet-induced downregulation of adipose HK2 contributes to hyperglycemia.

## Results

### HK2 is down-regulated in obese mice and humans

To determine the molecular basis of diet-induced hyperglycemia in mice, we performed an unbiased proteomic analysis on visceral white adipose tissue (vWAT) isolated from C57BL/6JRj wild-type mice fed a HFD for 4 weeks or normal diet (ND) (*Figure 1—figure supplement 1B–D*). We detected and quantified 6294 proteins of which 52 and 67 were up- and down-regulated, respectively, in vWAT of HFD-fed mice (*Supplementary file 1*). The glycolytic enzyme Hexokinase 2 (HK2), expressed in adipose tissue and muscle, was among the proteins significantly down-regulated in vWAT (*Figure 1A*). We focused on HK2 due to its role in glucose metabolism. HK2 phosphorylates glucose to generate glucose-6-phosphate (G6P), the rate-limiting step in glycolysis, in an insulin-stimulated manner. HK2 is the most abundant (~80%) of the three hexokinase isoforms expressed in vWAT but the only one down-regulated upon HFD (*Figure 1A* and *Figure 1—figure supplement 1E*). Quantification by immunoblotting revealed an approximately 60%, 90% and 50% reduction in HK2 expression in vWAT, subcutaneous WAT (sWAT), and brown adipose tissue (BAT), respectively, of 4 week HFD mice (*Figure 1B–C*). Consistent with reduced HK2 expression, hexokinase activity was decreased in WAT of HFD-fed mice (*Figure 1D*). HK2 expression was also decreased in WAT of ND-fed *ob/ob* mice, compared to littermate controls (*Figure 1—figure supplement 1F*), suggesting that loss of HK2 is common to different obesogenic conditions. A longitudinal study of HFD-fed mice revealed that HK2 down-regulation in adipose tissue occurred within one week of HFD and correlated with systemic insulin insensitivity (*Figure 1—figure supplement 2A–F*). HK2 expression was unchanged in skeletal muscle (*Figure 1B*). Expression of the glucose transporter GLUT4 was slightly reduced in vWAT at 4 weeks of HFD (*Figure 1A* and *Figure 1—figure supplement 2D*). In sWAT, GLUT4 was down-regulated at ≥1 week of HFD (*Figure 1—figure supplement 2E*). Confirming earlier observations (*Shimobayashi et al., 2018*; *Tan et al., 2015*), insulin signaling was normal in WAT at 4 weeks of HFD (*Figure 1—figure supplements 1A and 2D–E*). However, in BAT, insulin signaling was significantly downregulated within 1 week of HFD (*Figure 1—figure supplement 2F*). Shifting from HFD to 2 weeks of ND restored HK2 expression and normal blood glucose (*Figure 1E–F* and *Figure 1—figure supplement 2C*). Thus, loss of HK2 expression upon obesogenic conditions is an adipose-specific, transient physiological response that correlates with hyperglycemia.

Similar to our findings in mice, omental WAT (human vWAT) biopsies from obese non-diabetic and obese diabetic patients displayed a~30% reduction in hexokinase activity (*Figure 1G* and *Supplementary file 2*). Importantly, hexokinase activity was particularly low in obese non-diabetic patients

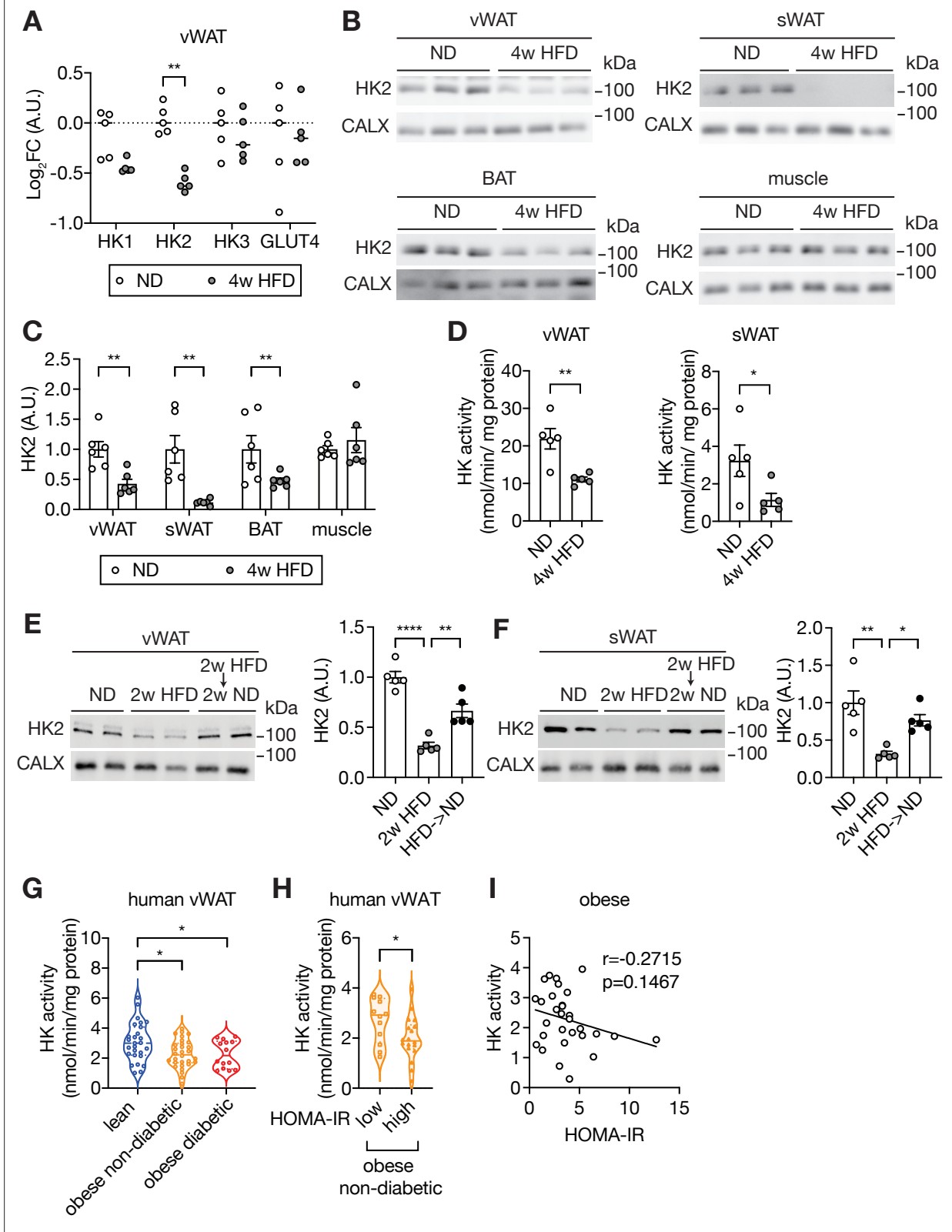

**Figure 1.** Loss of HK2 in obese mouse and obese human. (**A**) The $Log_2$ fold change (FC) of Hexokinase and GLUT4 protein expression in visceral white adipose tissue (vWAT) of normal diet (ND)- and 4 week high-fat diet (HFD)-fed wild-type C57BL/6JRj mice. Multiple t test, **q<0.0001. n=5 (ND) and 5 (HFD). (**B**) Immunoblot analyses of vWAT, subcutaneous WAT (sWAT), brown adipose tissue (BAT), and skeletal muscle from ND- and 4 week HFD-fed mice. CALX serves as a loading control. n=6 (ND) and 6 (HFD). (**C**) Quantification of panel B. Data is normalized to the loading control. Student's t test.

*Figure 1 continued on next page*

*Figure 1 continued*

**p<0.01. (**D**) Hexokinase (HK) activity of vWAT and sWAT from ND- and 4 week HFD-fed mice. Student's t test, *p<0.05, **p<0.01. n=5 (ND) and 5 (HFD). (**E–F**) Immunoblot analyses of vWAT (**E**) and sWAT (**F**) from ND-, 2 week HFD-, and 2 week HFD +2 week ND-fed mice. n=5 (ND), 5 (HFD), and 6 (HFD +ND). Data is normalized to the loading control. One-way ANOVA. *p<0.01, **p<0.01, ****p<0.0001. (**G**) Hexokinase (HK) activity of vWAT from lean, obese non-diabetic, and obese diabetic patients. Two-way ANOVA, *p<0.05. n=27 (lean), 30 (obese), and 14 (obese diabetic). (**H**) Comparison of vWAT HK activity from low or high HOMA-IR obese non-diabetic patients. Student's t tests, *p<0.05. n=12 (low, HOMA-IR <2.9) and 18 (high, HOMA-IR >2.9). (**I**) Pearson's correlation analyses of hexokinase activity and homeostatic assessment for insulin resistance (HOMA-IR) in obese patients. See *Figure 1—source data 1*.

The online version of this article includes the following source data and figure supplement(s) for figure 1:

**Source data 1.** Uncropped blots and source data for graphs for *Figure 1*.

**Figure supplement 1.** Supporting data 1 for *Figure 1*.

**Figure supplement 1—source data 1.** Uncropped blots and source data for graphs for *Figure 1—figure supplement 1*.

**Figure supplement 2.** Supporting data 2 for *Figure 1*.

**Figure supplement 2—source data 1.** Uncropped blots and source data for graphs for *Figure 1—figure supplement 2*.

with severe insulin resistance, compared to patients with mild insulin resistance (*Figure 1H*). Although hexokinase activity negatively correlated with insulin resistance in obese patients, this correlation was not statistically significant (*Figure 1I*), suggesting that loss of hexokinase activity may not be the only cause of insulin resistance in human. We note that Ducluzeau et al. observed a decrease in *HK2* mRNA expression in adipose tissue of diabetic patients (*Ducluzeau et al., 2001*). Altogether, the above findings suggest that HK2 down-regulation in adipose tissue is a key event, possibly causal, in obesity-induced insulin insensitivity and hyperglycemia in mouse and human.

## A loss-of-function *hk2* mutation in hyperglycemic cavefish

Mexican cavefish (*Astyanax mexicanus*), also known as blind fish, are hyperglycemic compared to surface fish from which they are descended (*Riddle et al., 2018*; *Figure 2A–B*). Hyperglycemia, although a pathological condition in mouse and human, is a selected trait that presumably allows cave-dwelling fish to survive in nutrient-limited conditions. Among three independently evolved cavefish isolates, Pachón and Tinaja cavefish (names refer to the caves from which the fish were isolated) acquired a loss-of-function mutation in the insulin receptor gene, causing insulin resistance and hyperglycemia (*Riddle et al., 2018*; *Figure 2B*). The third isolate, Molino cavefish, is the most hyperglycemic but contains a wild-type insulin receptor gene and displays normal insulin signaling (*Riddle et al., 2018*). Since the phenotype of Molino fish is similar to that of HFD-fed mice (normal insulin signaling yet hyperglycemic), we hypothesized that this cavefish may be hyperglycemic due to a loss-of-function mutation in the *hk2* gene. DNA sequencing of the *hk2* gene of surface fish and the three cavefish variants revealed a mutation in the *hk2* gene uniquely in Molino. The homozygous, missense mutation in the coding region of the Molino *hk2* gene changed highly conserved arginine 42 (R42) to histidine (R42H) (*Figure 2C and D*). Based on the published structure of HK2 (*Nawaz et al., 2018*), R42 forms a salt bridge with aspartic acid 272 (D272) to stabilize the conformation of HK2 (*Figure 2E*), predicting that R42H destabilizes HK2 and is thus a loss-of-function mutation. To test this prediction, we expressed surface fish and Molino HK2 in HEK293T cells which have low intrinsic hexokinase activity. Indeed, Molino HK2 displayed little-to-no hexokinase activity compared to surface fish HK2 (*Figure 2F*). To test whether R42H causes loss of function in mammalian HK2, we examined hexokinase activity of recombinant mammalian HK2 containing the Molino mutation (HK2-R42H). HK2-R42H displayed ~50% hexokinase activity compared to HK2-WT, despite similar expression levels (*Figure 2G*). Thus, the R42H mutation may account for the hyperglycemia in Molino cavefish. In other words, R42H in Molino appears to be a genetically fixed version of what we observe in mice as a physiological down-regulation of HK2 in response to HFD. Although further study is required to link the R42H mutation to the hyperglycemic phenotype in Molino cavefish, the findings in Molino provide orthogonal evidence that the down-regulation of HK2 in mice is physiologically relevant in hyperglycemia.

## Adipose-specific *Hk2* knockout causes hyperglycemia

The above findings altogether suggest that adipose-specific loss of HK2 may be a cause of hyperglycemia. To test this hypothesis, we first generated a stable HK2 knockdown pre-adipocyte 3T3-L1

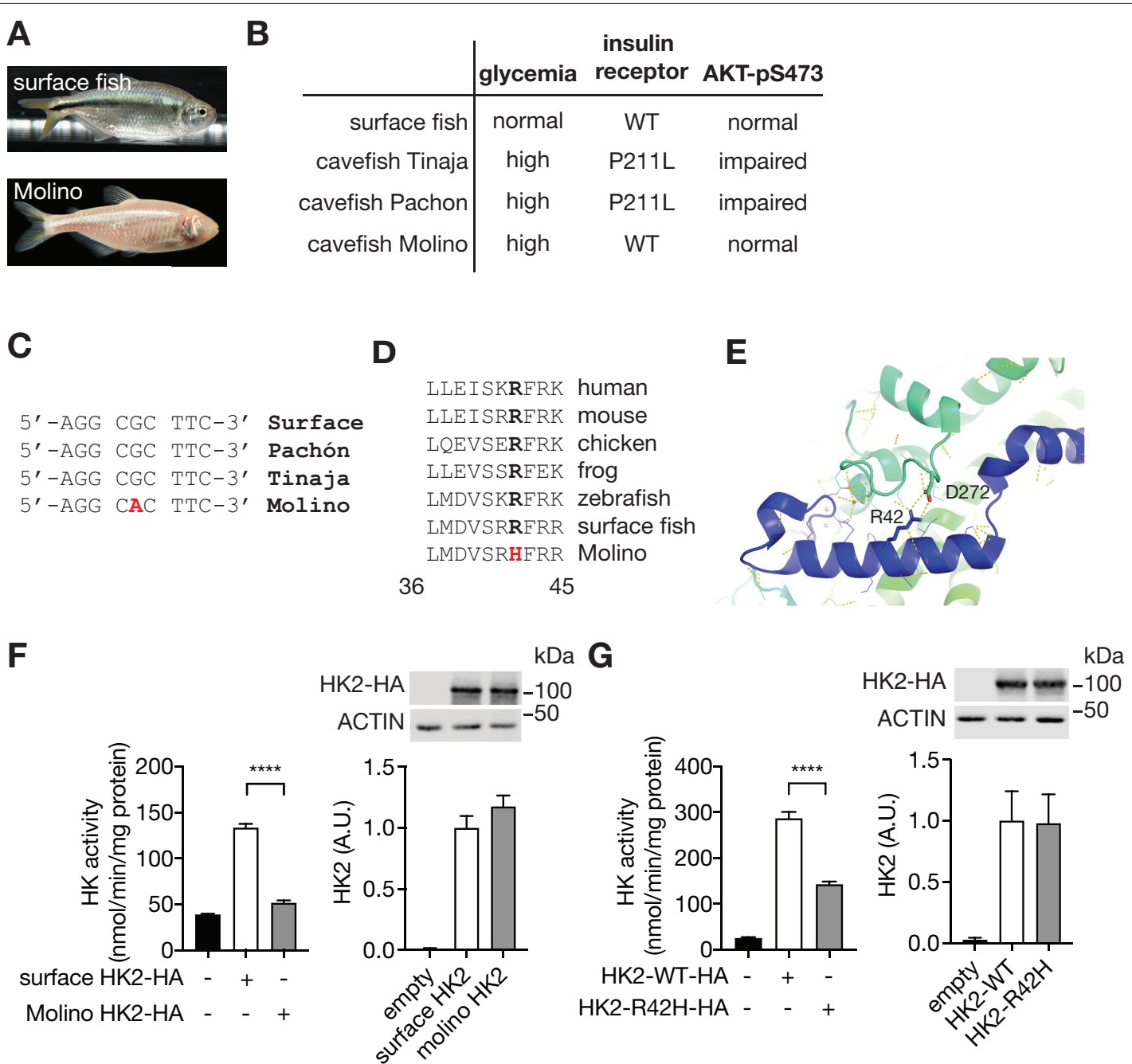

**Figure 2.** A loss-of-function HK2 variant in hyperglycemic Mexican cavefish. (**A**) Surface fish and Mexican cavefish Molino. (**B**) Comparisons of phenotypes in surface fish, Pachón, Tinaja, and Molino. (**C**) DNA sequence of the Molino variant. (**D**) Amino acid sequence alignment of the HK2-R42H mutation within vertebrates. (**E**) Structural analyses revealed the presence of a salt bridge between Arginine 42 (R42) and Aspartic acid 272 (D272) in the human HK2 (PDB: 2MTZ). (**F**) HK activity and immunoblot analyses for lysates of HEK293T cells expressing surface or Molino HK2. Student's t test, ****p<0.0001. N=4. (**G**) HK activity and immunoblots for lysates of HEK293T cells expressing control, HK2-WT, or HK2-R42H. Student's t test, ****p<0.0001. N=4. See *Figure 2—source data 1*.

The online version of this article includes the following source data for figure 2:

**Source data 1.** Uncropped blots and source data for graphs for *Figure 2*.

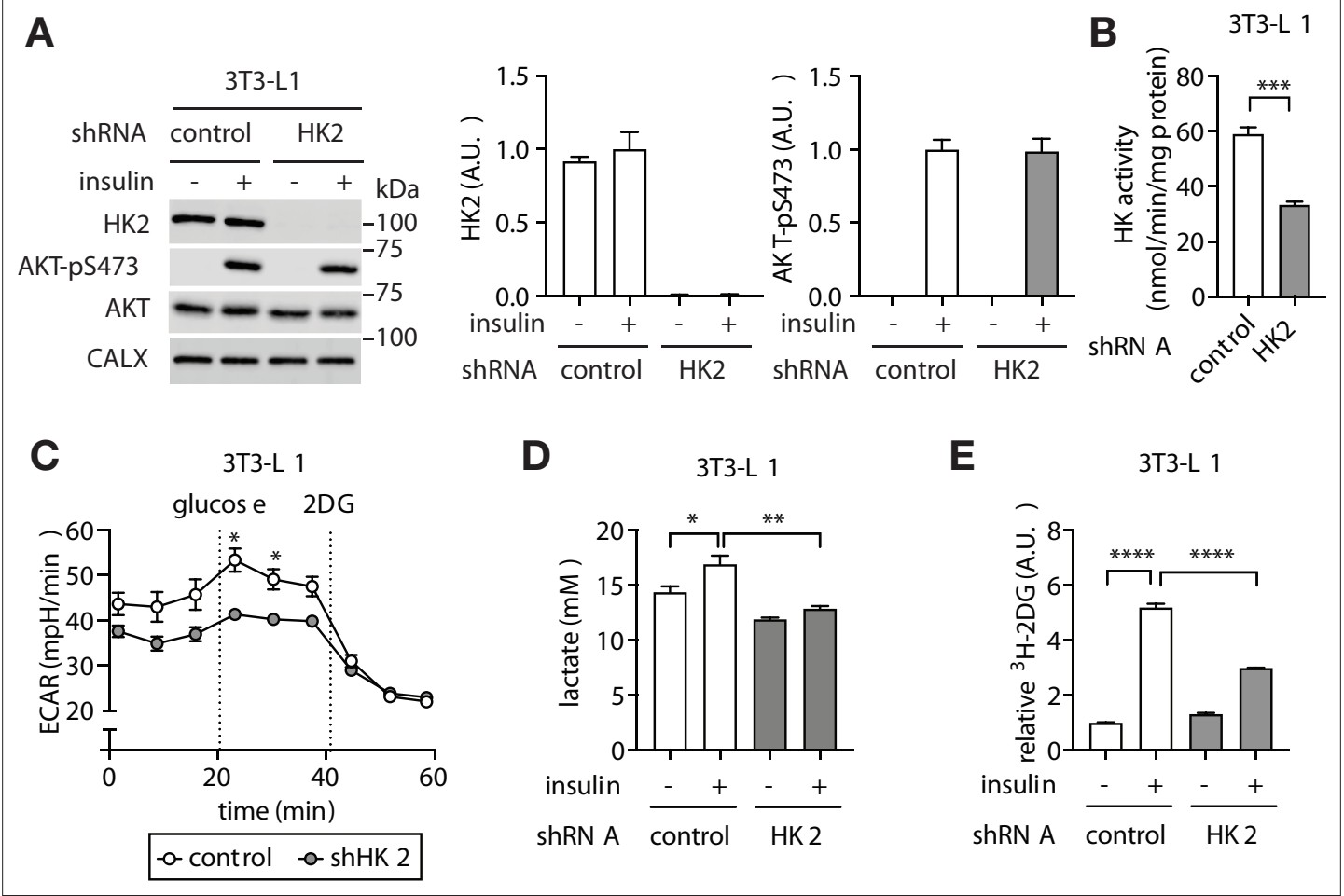

**Figure 3.** Loss of adipose HK2 causes reduced glucose disposal in adipocytes. (**A**) Immunoblot analyses of control and HK2 knockdown 3T3-L1 adipocyte lysates. Cells were stimulated with 100 nM insulin for 25 min. N=4. (**B**) HK activity of control and HK2 knockdown 3T3-L1 adipocytes. Student's t test, ***p<0.001. N=3. (**C**) Extracellular acidification rate of control and HK2 knockdown 3T3-L1 adipocytes in response to glucose (10 mM) and 2-deoxyglucose (2DG, 50 mM). Two-way ANOVA. *p<0.05. N=2. (**D**) Lactate secreted into media by control and HK2 knockdown 3T3-L1 adipocytes treated with or without 100 nM insulin. One-way ANOVA. *p<0.05, **p<0.01. N=3. (**E**) 2DG uptake in control and HK2 knockdown 3T3-L1 adipocytes treated with or without 100 nM insulin. One-way ANOVA. ****p<0.0001. N=3. See *Figure 3—source data 1*.

The online version of this article includes the following source data and figure supplement(s) for figure 3:

**Source data 1.** Uncropped blots and source data for graphs for *Figure 3*.

**Figure supplement 1.** Supporting data for *Figure 3*.

**Figure supplement 1—source data 1.** Source images and source data for graphs for *Figure 3—figure supplement 1*.

cell line (*Figure 3A–B*). HK2-knockdown pre-adipocytes differentiated normally to produce mature adipocytes (*Figure 3—figure supplement 1A–B*). The glycolytic rate was lower in HK2-knockdown adipocytes compared to controls, as measured by extracellular acidification rate (ECAR) (*Figure 3C*) and lactate production (*Figure 3D*). Furthermore, although basal glucose accumulation did not differ, insulin-stimulated glucose accumulation was 50% lower in HK2-knockdown adipocytes (*Figure 3E*), despite normal insulin signaling (*Figure 3A*). The observation that HK2-knockdown has no effect on basal glucose accumulation is due to the fact that HK2 is inactive in the absence of insulin (*Gottlob et al., 2001*; *Miyamoto et al., 2008*). Thus, loss of HK2 decreases glucose disposal in insulin-stimulated adipocytes in vitro.

To examine further the role of adipose HK2 in glucose homeostasis, and in particular the causality of HK2 loss in insulin insensitivity and hyperglycemia, we generated an adipose-specific *Hk2* knockout (AdHk2KO) mouse in which the knockout was induced at 4–5 weeks of age (*Figure 4—figure supplement 1A*). In AdHk2KO mice, HK2 expression was decreased ~75% in vWAT, ~70% in sWAT, and ~65%

in BAT but unchanged in skeletal muscle (*Figure 4A* and *Figure 4—figure supplement 1B–D*). Glycolytic metabolites were also decreased in WAT (*Figure 4B*). Furthermore, insulin signaling was normal in adipose tissue (*Figure 4A* and *Figure 4—figure supplement 1B–C*) and other tissues (see below) of AdHk2KO mice. ND-fed AdHk2KO mice displayed slightly less fat mass and slightly more lean mass than controls, but no difference in overall body weight (*Figure 4—figure supplement 2A–B*). No significant difference was observed in the weight of individual organs except for a small decrease in vWAT and a small increase in liver weight (*Figure 4—figure supplement 2C–D*). vWAT and sWAT from AdHk2KO and control mice were morphologically indistinguishable (*Figure 4—figure supplement 2E*). AdHk2KO and control mice displayed similar mRNA expression and circulating levels of leptin and adiponectin (*Figure 4—figure supplement 2F–I*). Expression of brown adipocyte markers (*Ucp1* and *Dio2*) was also unchanged in AdHk2KO mice (*Figure 4—figure supplement 2J*). The above similarities in adipose tissue from AdHk2KO and control mice are not surprising given that the HK2 knockout was induced at 4–5 weeks of age, after adipose tissue is fully developed. However, contrary to controls, ND-fed AdHk2KO mice were insulin insensitive and severely glucose intolerant as measured by conventional tolerance tests (*Figure 4C–D*). Moreover, glucose intolerance was observed in adipose-specific heterozygous *Hk2* knockout mice (*Figure 4—figure supplement 3A–C*). These data indicate that partial loss of HK2, as observed in HFD-fed wild-type mice (*Figure 1*), is sufficient to cause glucose intolerance. We also measured the insulin insensitivity of AdHk2KO mice in hyperinsulinemic-euglycemic clamp conditions (*Figure 4E–F* and *Figure 4—figure supplement 3D–F*). Compared to control mice, AdHk2KO mice required significantly less or no infusion of glucose to maintain euglycemia (*Figure 4E–F*), despite similar hyperinsulinemia (*Figure 4—figure supplement 3D*). We note that the glycolytic rate before insulin infusion is higher in AdHk2KO mice (*Figure 4—figure supplement 3F*). The reason for this increased basal glycolytic rate is unknown. The above findings confirm that loss of HK2 specifically in adipose tissue is sufficient to cause insulin insensitivity and thereby hyperglycemia, even in ND-fed mice.

We also investigated HFD-fed AdHk2KO mice. No significant difference was observed in fat mass, body weight, organ weight and insulin tolerance in AdHk2KO mice compared to controls, in mice fed a HFD for 12 weeks (*Figure 4—figure supplement 4A–D*). The similar phenotype of HFD-fed AdHk2KO and control mice was expected due to the HFD-induced, early loss of HK2 expression in the control mice (*Figure 1A–B* and *Figure 1—figure supplement 2D–F*).

## Adipose-specific *Hk2* knockout causes selective insulin resistance in liver

Consistent with reduced glucose accumulation observed in adipocytes in vitro (*Figure 3E*), we also observed reduced glucose uptake in adipose tissue in vivo, as measured upon injection of the glucose tracer $^{14}$C-2-deoxyglucose during our hyperinsulinemic-euglycemic clamp studies (*Figure 5A*). However, adipose tissue accounts for only ≤5% of glucose disposal (*Jackson et al., 1986*; *Kowalski and Bruce, 2014*). Thus, the insulin insensitivity and glucose intolerance observed in AdHk2KO mice (*Figure 4C–F*) cannot be explained solely by impaired glucose uptake in adipose tissue. Glucose uptake in skeletal muscle, which accounts for 70–80% of insulin-inducible glucose clearance (*Kowalski and Bruce, 2014*), was unimpaired in AdHk2KO mice (*Figure 5A*). Plasma insulin levels were also similar in AdHk2KO and control mice (*Figure 5—figure supplement 1A–B*). Moreover, insulin signaling was not affected in skeletal muscle and liver of fasted and re-fed AdHk2KO mice (*Figure 4—figure supplement 1D* and *Figure 5—figure supplement 1C*). These findings indicate that the severity of the glucose intolerance in AdHk2KO mice is not due to defects in glucose uptake in skeletal muscle, insulin secretion or insulin signaling in peripheral tissues.

Another potential explanation for the severity of glucose intolerance in AdHk2KO mice is de-repression of hepatic glucose production. Adipose tissue is known to impinge negatively on glucose production in liver (*Abel et al., 2001*; *Kumar et al., 2010*; *Tang et al., 2016*; *Vijayakumar et al., 2017*), the main glucose producing organ. We indeed observed increased glucose production in AdHk2KO mice under hyperinsulinemic-euglycemic clamp conditions (*Figure 5B*). To test further whether the increased glucose production in AdHk2KO mice is due to enhanced gluconeogenesis, we performed a pyruvate tolerance test (PTT). AdHk2KO mice displayed significantly higher production of glucose upon pyruvate injection, compared to controls (*Figure 5C*). Consistent with the observed increase in pyruvate-dependent glucose production, gluconeogenic genes (*G6pc* and *Pepck*) were

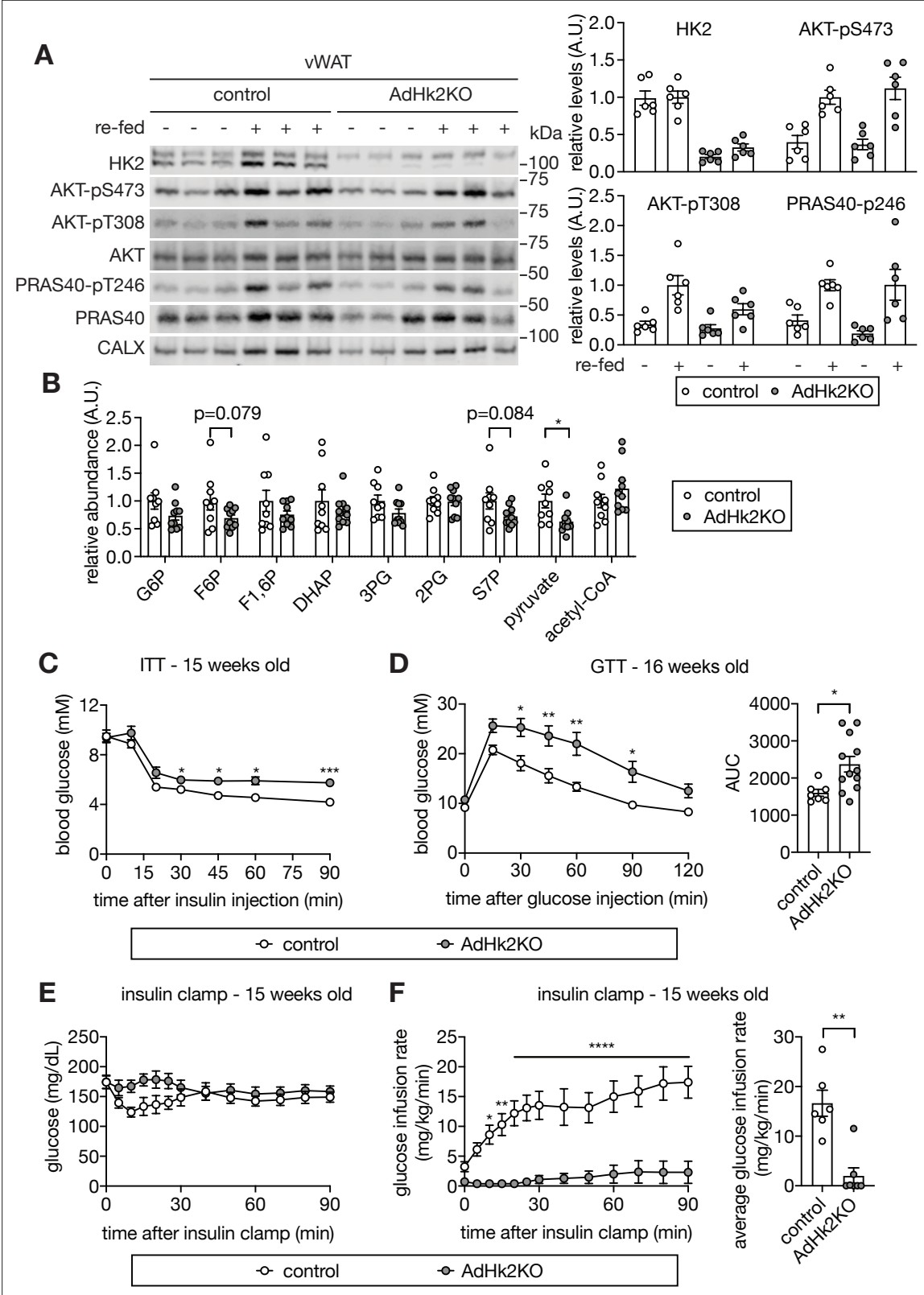

**Figure 4.** Loss of adipose HK2 causes hyperglycemia in mice. (**A**) Immunoblot analyses of HK2 expression and insulin signaling in vWAT of control and adipose-specific Hk2 knockout (AdHk2KO) mice. Mice were fasted overnight or fasted overnight and re-fed for 3 hours. Quantification data is normalized to a loading control. n=6. (**B**) Fold change (AdHk2KO/control) of metabolites in glycolysis and the pentose phosphate pathway in vWAT of control and AdHk2KO mice. Student's t test. *p<0.05. n=9 (control) and 10 (AdHk2KO). (**C**) Insulin tolerance test (ITT) on control and AdHk2KO mice. The

*Figure 4 continued on next page*

*Figure 4 continued*

mice were fasted for 6 hours and injected with insulin (0.5 U/kg body weight). Two-way ANOVA. *p<0.05, ***p<0.001. n=6 (control) and 10 (AdHk2KO). (**D**) Glucose tolerance test (GTT) on control and AdHk2KO mice. Mice were fasted for 6 hours and injected with glucose (2 g/kg body weight). Two-way ANOVA for glucose curves and Student's t test for AUC. *p<0.05, **p<0.01. n=7 (control) and 12 (AdHk2KO). (**E–F**) Hyperinsulinemic-euglycemic clamp studies on control and AdHk2KO mice. Mice were fasted for 6 hours. Under insulin clamp, euglycemia was maintained (**E**) by manipulating glucose infusion rate (**F**). Bar graph shows average glucose infusion rate under euglycemia. Two-way ANOVA for glucose infusion rate curve and Student's t test for average glucose infusion rate, **p<0.01, ****p<0.0001. n=6 (control) and 7 (AdHk2KO). See *Figure 4—source data 1*.

The online version of this article includes the following source data and figure supplement(s) for figure 4:

**Source data 1.** Uncropped blots and source data for graphs for *Figure 4*.

**Figure supplement 1.** Supporting data 1 for *Figure 4*.

**Figure supplement 1—source data 1.** Uncropped blots and source data for graphs for *Figure 4—figure supplement 1*.

**Figure supplement 2.** Supporting data 2 for *Figure 4*.

**Figure supplement 2—source data 1.** Source images and source data for graphs for *Figure 4—figure supplement 2*.

**Figure supplement 3.** Supporting data 3 for figure 4.

**Figure supplement 3—source data 1.** Uncropped blots and source data for graphs for *Figure 4—figure supplement 3*.

**Figure supplement 4.** Supporting data 4 for figure 4.

**Figure supplement 4—source data 1.** Source data for graphs for *Figure 4—figure supplement 4*.

---

upregulated in AdHk2KO liver (*Figure 5D*). These findings suggest that enhanced hepatic glucose production accounts for the severity of glucose intolerance in AdHk2KO mice. Thus, loss of adipose HK2 non-cell-autonomously promotes glucose intolerance by enhancing hepatic glucose production.

We observed enhanced glucose production in AdHk2KO mice despite normal systemic insulin signaling, including in liver (*Figure 5—figure supplement 1C*). This is an apparent paradox because insulin signaling normally inhibits glucose production. To investigate this paradox, we examined hepatic lipogenic gene expression, another readout for insulin action. The insulin-stimulated transcription factors sterol regulatory element-binding protein 1 c (SREBP1c) and carbohydrate responsive element binding protein (ChREBP) activate fatty acid synthesis genes and thereby promote de novo lipogenesis in liver (*Horton et al., 2002*; *Iizuka et al., 2004*). Expression of *Srebp1c*, *Mlxipl/Chrebp*, and fatty acid synthesis genes (*Acly*, *Scd1*, *Fasn*, and *Acc*) was maintained, even mildly increased, in liver of AdHk2KO mice (*Figure 5D*). FASN and ACC protein levels were also slightly increased in liver of AdHk2KO mice (*Figure 5—figure supplement 1C*). These findings indicate that loss of adipose HK2 does not inhibit hepatic lipogenesis although hepatic glucose production is enhanced. The increased hepatic glucose production and maintained lipogenesis, as observed in AdHk2KO mice, resembles a condition in diabetic patients known as selective insulin resistance (*Brown and Goldstein, 2008*; see below). Thus, loss of adipose HK2 causes selective insulin resistance and thereby contributes to the pathogenesis of type 2 diabetes.

We note that hepatic triglyceride (TG) levels were unchanged in AdHk2KO mice despite increased expression of lipogenic genes and enzymes in liver (*Figure 5D* and *Figure 5—figure supplement 1C–E*) and increased de novo TG synthesis (*Figure 5E*). Most likely, de novo-synthesized hepatic TG in AdHk2KO mice is secreted and delivered to adipose tissue for storage, as suggested by the higher levels of circulating TG and cholesterol in AdHk2KO mice (*Figure 5F–G*).

## Adipose-specific *Hk2* knockout impairs lipogenesis and enhances fatty acid release in adipose tissue

How does loss of HK2 in adipose tissue increase hepatic glucose production? Adipose lipogenesis non-cell-autonomously inhibits hepatic glucose production (*Vijayakumar et al., 2017*). Adipose-specific knockout of the transcription factor ChREBP decreases adipose lipogenesis and thereby increases hepatic glucose production (*Ortega-Prieto and Postic, 2019*; *Vijayakumar et al., 2017*). Glucose-derived metabolites have been shown to promote ChREBP activity (*Dentin et al., 2012*; *Kabashima et al., 2003*; *Kawaguchi et al., 2002*; *Kim et al., 2016*; *Li et al., 2010*). Based on these observations, we hypothesized that HFD-induced loss of ChREBP activity and thus lipogenesis in adipose tissue may cause increased hepatic glucose production. To test this hypothesis, we examined *Mlxipl/Chrebp* expression in adipose tissue of AdHk2KO mice. ChREBP has two isoforms. Constitutively

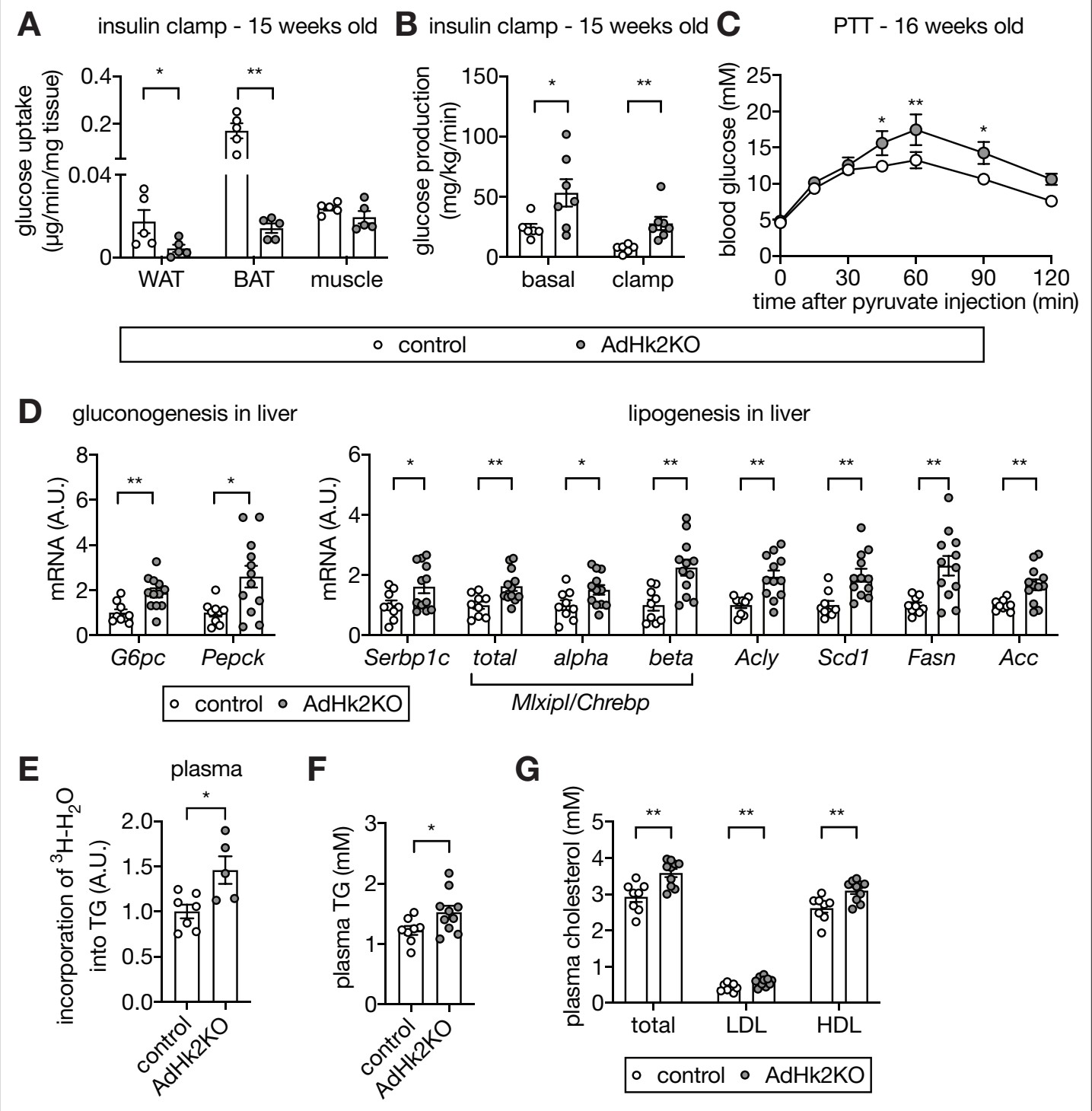

**Figure 5.** Loss of adipose HK2 causes reduced glucose disposal in adipose tissue and de-represses glucose production in liver. (**A**) Glucose uptake measured at the end of hyperinsulinemic-euglycemic clamp. 2DG was injected 30 min prior to organ collections. The 2DG values were normalized by tissue mass used for the assay. Mann-Whitney test, *p<0.05, **p<0.01. n=5 (control) and 5 (AdHk2KO) mice. (**B**) Endogenous glucose production under basal and hyperinsulinemic-euglycemic clamp conditions. Mann-Whitney test, *p<0.05, **p<0.01. n=6 (control) and 7 (AdHk2KO) mice. (**C**) Pyruvate tolerance test (PTT) on control and AdHk2KO mice. Mice were fasted for 15 hours and injected with pyruvate (2 g/kg body weight). Two-way ANOVA, *p<0.05, **p<0.01. n=5 (control) and 6 (AdHk2KO). (**D**) mRNA levels of gluconeogenic genes (left) and lipogenic genes (rignt) in liver of control and AdHk2KO mice. Multiple t test, *p<0.05, **p<0.01. n=9 (control) and 12 (AdHk2KO). (**E**) De novo-synthesized plasma TG. Mice were treated with $^3$H-$H_2O$ and incorporation of $^3$H in plasma TG was measured. Student's t test, *p<0.05. n=7 (control) and 5 (AdHk2KO). (**F**) Plasma triglyceride (TG) levels

*Figure 5 continued on next page*

*Figure 5 continued*

in control and AdHk2KO mice. Student's t test, *p<0.05. n=8 (control) and 10 (AdHk2KO). (**G**) Plasma cholesterol, low density lipoprotein (LDL), or high density lipoprotein (HDL) levels in control and AdHk2KO mice. Multiple t test, **p<0.01. n=8 (control) and 10 (AdHk2KO). See *Figure 5—source data 1*.

The online version of this article includes the following source data and figure supplement(s) for figure 5:

**Source data 1.** Source data for graphs for *Figure 5*.

**Figure supplement 1.** Supporting data 1 for figure 5.

**Figure supplement 1—source data 1.** Uncropped immunoblots, source images and source data for graphs for *Figure 5—figure supplement 1*.

expressed ChREBPα promotes transcription of ChREBPβ which then activates lipogenic genes (*Herman et al., 2012*). Expression of *Mlxipl-beta/Chrebp-beta,* but not *Mlxipl-alpha/Chrebp-alpha*, was significantly decreased in adipose tissue of AdHk2KO mice (*Figure 6A*). Consistent with reduced *Mlxipl-beta/Chrebp-beta* expression, lipogenic genes and enzymes and ultimately lipogenesis were down-regulated in adipose tissue of AdHk2KO mice (*Figure 6A–D*). The above findings indicate that loss of adipose HK2 causes loss of ChREBP activity and lipogenesis in adipose tissue. Furthermore, our above findings combined with previous literature (*Vijayakumar et al., 2017*) suggest that loss of adipose HK2 promotes hepatic glucose production via loss of adipose lipogenesis. We note that adipose tissue weight in AdHk2KO mice is essentially unchanged (*Figure 4—figure supplement 2C*) despite loss of lipogenesis, likely due to a compensating supply of TG from liver (*Figure 5D–F*).

Non-esterified fatty acid (NEFA) released from adipose tissue, in addition to loss of adipose lipogenesis, promotes glucose production in liver (*Perry et al., 2015*; *Titchenell et al., 2016*). Furthermore, ChREBP in adipose tissue both promotes lipogenesis and inhibits NEFA release (*Vijayakumar et al., 2017*). Thus, we examined the effect of HK2 loss on NEFA release. Ex vivo, insulin inhibited isoproterenol-induced NEFA release in adipose tissue from control mice, but not in explants from AdHk2KO mice (*Figure 6E*). In vivo, glucose administration inhibited NEFA release only in control mice (*Figure 6F–G*) despite similar physiological increases insulin levels in fasted AdHk2KO and control mice (*Figure 5—figure supplement 1B*), in agreement with the above ex vivo experiment. These findings suggest that loss of HK2 simultaneously inhibits lipogenesis and promotes NEFA release in adipose tissue, both of which contribute to de-repression of hepatic glucose production.

## Mechanisms of HFD-induced HK2 down-regulation

Our findings suggest that HFD causes loss of adipose HK2 and thereby triggers hyperglycemia. How does HFD down-regulate adipose HK2? To answer this question, we first examined *Hk2* mRNA expression in HFD- and ND-fed mice. HFD-fed mice exhibited decreased *Hk2* mRNA in sWAT but not in vWAT or BAT (*Figure 7A*). This suggests that HK2 synthesis is down-regulated at the transcriptional level in sWAT, and at a post-transcriptional level in vWAT and BAT. To investigate the post-transcriptional mechanism, we measured synthesis of HK2 in vWAT of HFD-fed mice. vWAT explants from HFD- or ND-fed mice were treated with L-azidohomoalanine (AHA), a methionine analog, and AHA-containing polypeptides were purified and quantified by mass spectrometry (*Supplementary file 3*). The amount of AHA-containing HK2 polypeptides was significantly decreased in vWAT explants from HFD-fed mice (*Figure 7B*). Thus, HFD down-regulates HK2 in vWAT by inhibiting *Hk2* mRNA translation.

Are there any other adipose proteins post-transcriptionally regulated like HK2? In our AHA-proteomics dataset (*Supplementary file 3*), we identified 155 proteins whose AHA incorporation positively correlated with HK2 AHA-incorporation. Pathway enrichment analysis of the 155 proteins yielded ribosomal proteins, the electron transport chain (ETC), oxidative phosphorylation (OXPHOS), fatty acid synthesis, and glycolysis as the top 5 pathways (*Figure 7—figure supplement 1A–D*). As demonstrated in *Figure 6*, downregulation of enzymes in fatty acid synthesis (e.g. FASN and ACC) was mainly due to reduced transcription (*Figure 7—figure supplement 1C*). However, reduced AHA-incorporation in ribosomal, ETC, and OXPHOS proteins was post-transcriptional (*Figure 7—figure supplement 1B*), suggesting that these proteins might be regulated in a manner similar to HK2. Regulated protein synthesis in adipose tissue may play an important role in the development of hyperglycemia.

It was recently reported that the forkhead transcription factors FOXK1 and FOXK2 promote *Hk2* transcription in adipose tissue (*Sukonina et al., 2019*). We investigated whether the transcriptional down-regulation of *Hk2* in sWAT is due to loss of FOXK1 and FOXK2. *Foxk1* and *Foxk2* expression

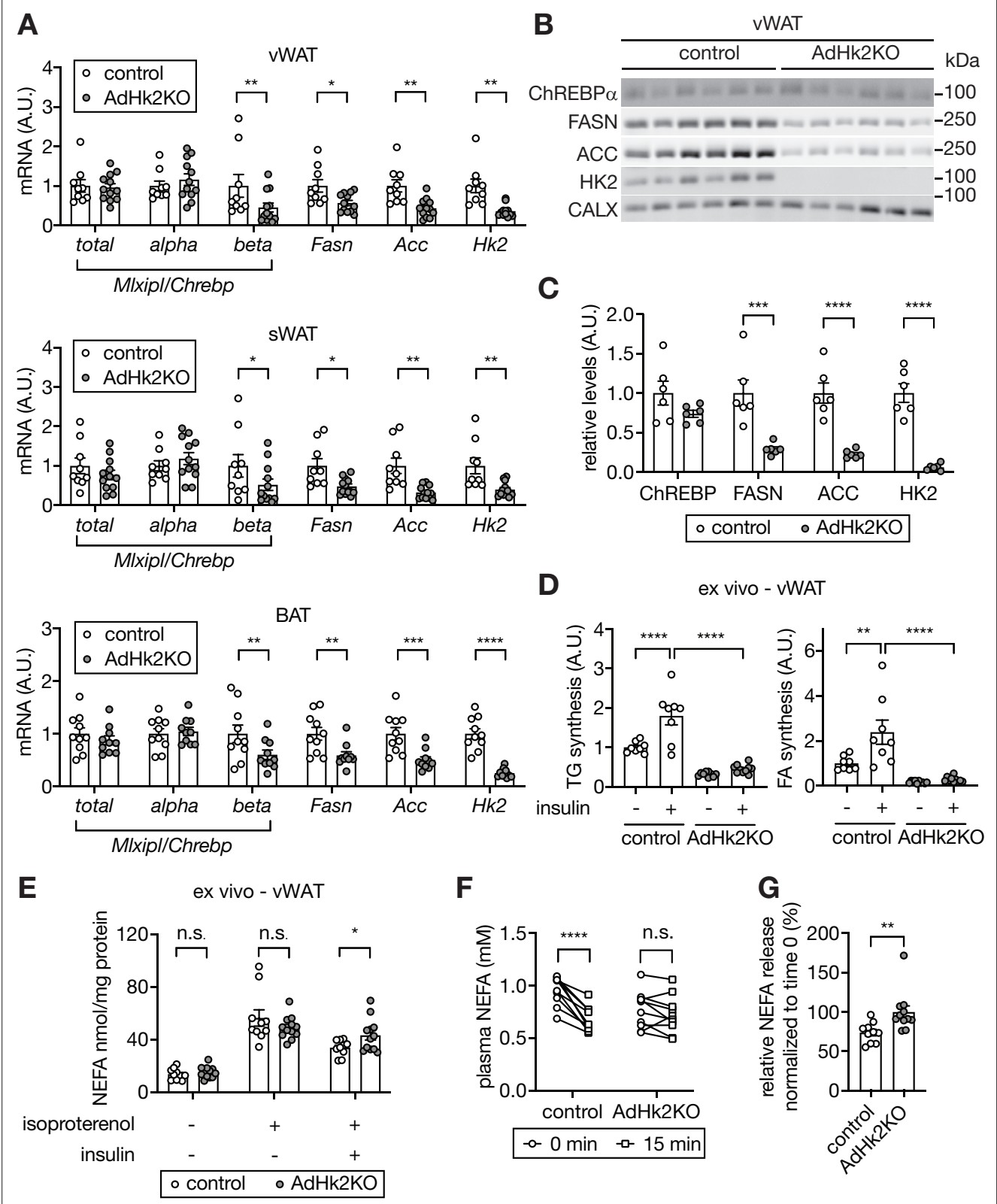

**Figure 6.** Decreased lipogenesis and enhanced fatty acid release in adipose tissue in AdHk2KO mice. (**A**) mRNA levels of fatty acid synthesis genes in vWAT (top), sWAT(middle), and BAT (bottom) from control and AdHk2KO mice. Multiple t test, *p<0.05, **p<0.01, ***p<0.001, ****p<0.0001. n=9 (control) and 12 (AdHk2KO) for vWAT and sWAT. n=10 (control) and 10 (AdHk2KO) for BAT. (**B**) Immunoblot analyses of fatty acid synthesis enzymes in vWAT of control and Adk2KO mice. n=6. (**C**) Quantification of panel B. Data is normalized to loading controls. Student's t test. **p<0.01. ****p<0.0001.

*Figure 6 continued on next page*

*Figure 6 continued*

(**D**) De novo lipogenesis of vWAT explants of control and AdHk2KO mice. vWAT explants were treated with $^3$H-H$_2$O in the absence or presence of 100 nM insulin for 1 hour. Two-way ANOVA, **p<0.01, ****p<0.0001. n=8 (control) and 10 (AdHk2KO). (**E**) Non-esterified fatty acid (NEFA) release of vWAT explants of control and AdHk2KO mice. vWAT explants were treated with or without 10 µM isoproterenol in the absence or presence of 100 nM insulin. Multiple t test, *p<0.05. n.s., not significant. n=10 (control) and 12 (AdHk2KO). (**F**) Plasma NEFA levels in control and AdHk2KO mice before (0 min) and after (15 min) glucose injection. Mice were fasted for 6 hours and injected with glucose (2 g/kg body weight). Two-way ANOVA, **p<0.001. n.s., not significant. n=10 (control) and 11 (AdHk2KO). (**G**) Plasma NEFA levels in control and AdHk2KO mice after (15 min) glucose injection was normalized by plasma NEFA levels at 0 min (before glucose injection) in E. Student's t test, **p<0.01. n=10 (control) and 11 (AdHk2KO). See *Figure 6— source data 1*.

The online version of this article includes the following source data for figure 6:

**Source data 1.** Uncropped blots and source data for *Figure 6*.

was significantly decreased in sWAT, but not in vWAT or BAT, of HFD-fed mice (*Figure 7C–E*). Thus, HFD appears to down-regulate HK2 in sWAT by inhibiting FOXK1/2 and, thereby, expression of the *Hk2* gene.

## Discussion

How obesity causes insulin insensitivity and hyperglycemia is a long-standing question. In this study, we show that HFD causes hyperglycemia by inducing loss of HK2 specifically in adipose tissue. Loss of adipose HK2 is sufficient to cause glucose intolerance, in two ways (*Figure 8*). First, HK2 loss results in reduced glucose disposal by adipose tissue due to the inability of adipocytes to trap and metabolize non-phosphorylated glucose (*Figure 5A*). Second, loss of HK2 in adipocytes de-represses glucose production in liver (*Figure 5B*). We propose that loss of adipose HK2 is a mechanism of diet-induced insulin insensitivity and hyperglycemia.

In liver, insulin normally represses glucose production to decrease blood glucose and up-regulates lipogenesis to increase energy storage. Paradoxically, in type 2 diabetes, insulin fails to inhibit glucose production yet stimulates lipogenesis, hence the liver is selectively insulin resistant (*Brown and Goldstein, 2008*). Furthermore, previous findings suggest that selective insulin resistance occurs despite intact hepatic insulin signaling (*Titchenell et al., 2016*). Consistent with these previous findings, we observed that loss of adipose HK2 causes selective insulin resistance without affecting hepatic insulin signaling. Thus, we propose that loss of adipose HK2 is a mechanism for selective insulin resistance in liver and ultimately diabetes.

How does adipose HK2 control hepatic glucose production? As described above, one possible explanation is that decreased lipogenesis and increased NEFA release in adipose tissue, due to loss of HK2, promotes glucose production (*Figure 6*; *Perry et al., 2015*; *Titchenell et al., 2016*; *Vijayakumar et al., 2017*). Another interesting possibility is that loss of adipose HK2 causes increased hepatic glucose production via the central nervous system (CNS). Adipose tissue has a sensory nervous system (*Blaszkiewicz et al., 2019*; *Fishman and Dark, 1987*; *Frei et al., 2022*; *Kreier et al., 2006*; *Makwana et al., 2021*; *Song et al., 2009*; *Wang et al., 2022*) which may communicate with the liver via the sympathetic nervous system. Sympathetic activity promotes hepatic glucose production (*Niijima and Fukuda, 1973*; *Shimazu and Fukuda, 1965*). The above two models are not mutually exclusive since released bioactive fatty acids could act directly on the liver and/or adipose sensory neurons. A third possibility to explain increased hepatic glucose production is that increased glucose uptake by the liver due to reduced glucose disposal in adipose tissue may stimulate hepatic glucose production, consistent with the demonstration by Kim et al. that hepatic ChREBP promotes glucose production (*Kim et al., 2016*).

The phenotype of AdHk2KO mice is similar to that of adipose-specific GLUT4 knockout mice (*Abel et al., 2001*). Both knockout mice display reduced lipogenesis in adipose tissue and are hyperglycemic due to decreased glucose disposal in adipose tissue and increased glucose production in the liver (*Vijayakumar et al., 2017*). The phenotypic similarity of GLUT4 and HK2 knockout mice underscores the importance of adipose glucose metabolism in systemic insulin sensitivity and glucose homeostasis. However, our findings on diet-induced HK2 downregulation provide important new insight on the development of diet-induced hyperglycemia. First, unlike GLUT4, HK2 downregulation in adipose tissue correlates with HFD-induced hyperglycemia in mice (*Figure 1—figure supplement 2*). Second,

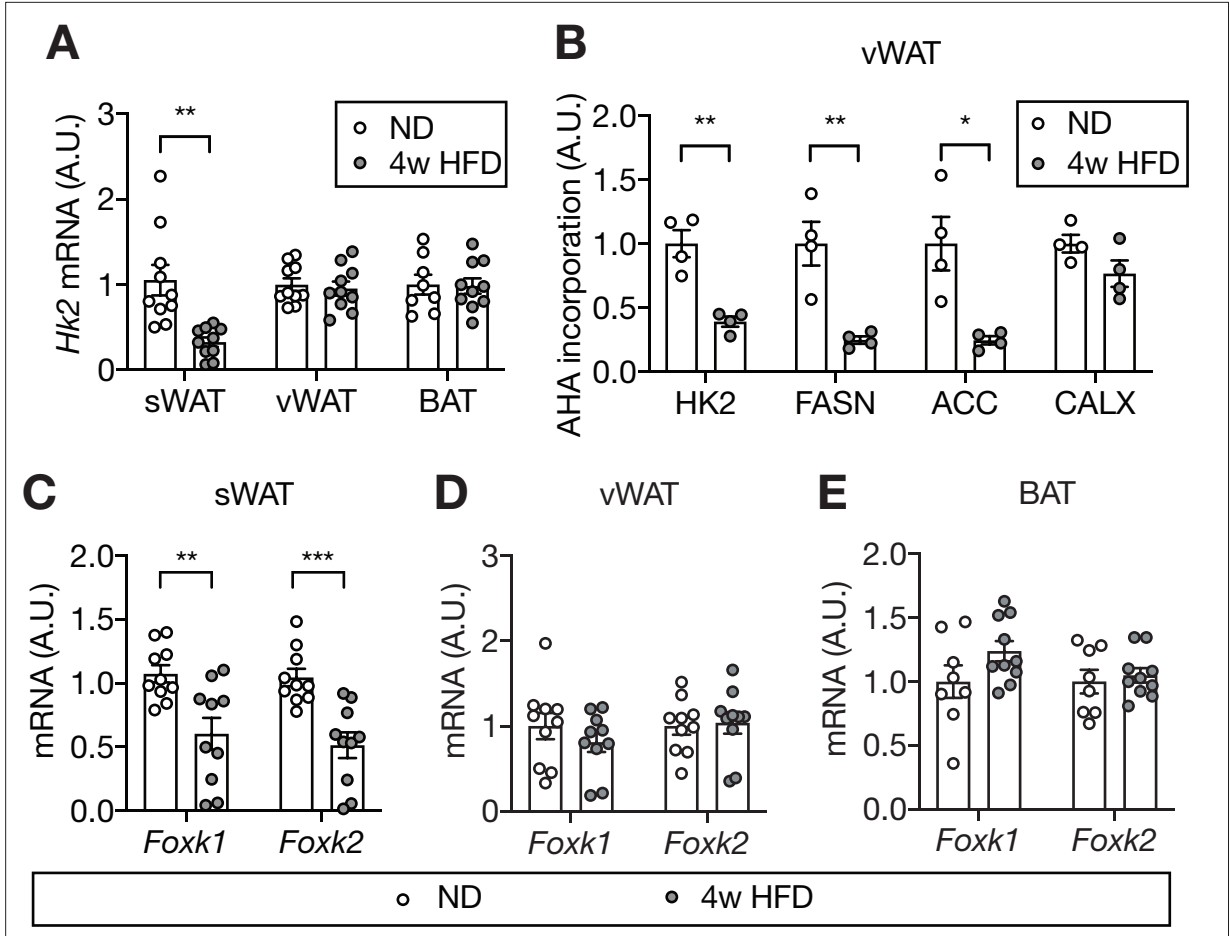

**Figure 7.** Mechanism of HFD-induced HK2 down-regulation in adipose tissue. (**A**) *Hk2* mRNA levels in sWAT, vWAT, and BAT from ND- and 4 week HFD-fed mice. Student's t test, **p<0.01. n=10 (sWAT from ND and HFD), 10 (vWAT from ND and HFD), 8 (BAT from ND), 10 (BAT from HFD). (**B**) Nascent polypeptides. vWATs isolated from ND- or 4 week HFD-fed mice were labled with L-azidohomoalanine (AHA), and AHA-incorporated polypeptides were measured by mass spectrometer. FASN and ACC serve as positive controls and CALX serves as a negative control. Multiple t test, *p<0.05, **p<0.01. n=4 (ND) and 4 (HFD). (**C**) *Foxk1* and *Foxk2* mRNA levels in sWAT from ND- or 4 week HFD-fed mice. Multiple t test, **p<0.01, ***p<0.001. n=10 (ND) and 10 (HFD). (**D**) *Foxk1* and *Foxk2* mRNA levels in vWAT of ND- or 4 week HFD-fed wild-type C57BL6JRj mice. No significant difference in multiple t test. n=10 (ND) and 10 (HFD). (**E**) *Foxk1* and *Foxk2* mRNA levels in BAT of ND- or 4 week HFD-fed wild-type C57BL6JRj mice. No significant difference in multiple t test. n=8 (ND) and 10 (HFD). See *Figure 7—source data 1*.

The online version of this article includes the following source data and figure supplement(s) for figure 7:

**Source data 1.** Source data for graphs for *Figure 7*.

**Figure supplement 1.** Transcriptionally and post-transcriptionally de-regulated proteins in vWAT of HFD-fed mice.

**Figure supplement 1—source data 1.** Source data for graphs for *Figure 7—figure supplement 1*.

it has been shown that glucose or a downstream metabolite(s) promotes ChREBP activity and lipogenesis in metabolic organs including adipose tissue (*Ortega-Prieto and Postic, 2019*). Two previous studies demonstrated that HFD prevents ChREBP activation and lipogenesis in adipose tissue, which leads to hyperglycemia due to enhanced hepatic glucose production in mice (*Herman et al., 2012*; *Vijayakumar et al., 2017*). Importantly, one of these studies showed that HFD inhibits ChREBP activity in adipose tissue independently of GLUT4 expression (*Herman et al., 2012*), concluding that the underlying mechanism of HFD-induced inhibition of ChREBP and hyperglycemia is still unknown. Our findings suggest that the physiological mechanism of HFD-induced inhibition of adipose ChREBP and lipogenesis, and consequently hyperglycemia, may be loss of HK2. Thus, we propose that the primary defect in adipose tissue of HFD-fed mice is not at the level of GLUT4 expression and glucose transport, but rather at the level of HK2 expression and glucose phosphorylation.

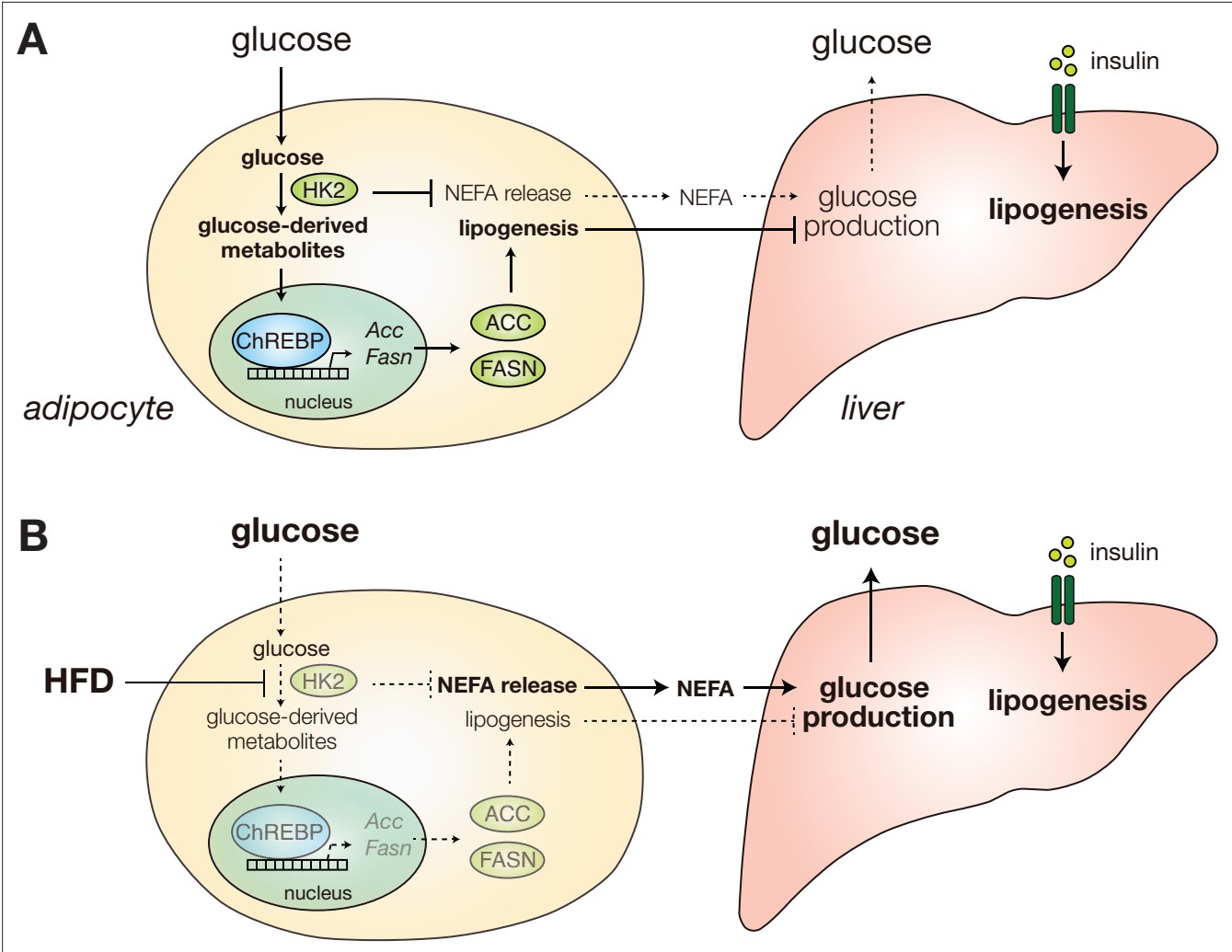

**Figure 8.** Diet-induced HK2 in adipose tissue promotes glucose intolerance. (**A**) HK2 promotes lipogenesis and suppresses NEFA release in adipose tissue (left), suppressing hepatic glucose production (right) and thus maintaining glucose homeostasis. (**B**) Diet-induced loss of adipose HK2 triggers glucose intolerance via reduced glucose disposal in adipocytes (left) and de-repressed hepatic gluconeogenesis despite maintained lipogenesis (right).

Can HFD-resistant HK2 expression in adipose tissue prevent diet-induced insulin insensitivity and hyperglycemia? To this point, our attempts to express HK2 in WAT of HFD-fed mice were unsuccessful (unpublished data), most likely because HK2 expression is tightly controlled at the post-transcriptional level (*Figure 7*). Importantly, expression of glucokinase (also known as HK4, hepatic hexokinase) in adipose tissue has been shown to prevent insulin insensitivity in HFD-fed mice (*Muñoz et al., 2010*). Elucidating the molecular mechanism of HFD-induced translational repression of HK2 may lead to a novel strategy in the treatment of insulin insensitivity and type 2 diabetes.

A previous study demonstrated that global heterozygous *Hk2* knockout mice have normal, even better, glucose tolerance compared to controls (*Heikkinen et al., 1999*). How can one explain this apparent discrepancy between global *Hk2* knockout and our adipose-specific *Hk2* knockout? We note that adipose-specific GLUT4 knockout mice display glucose intolerance (*Abel et al., 2001*), but global GLUT4 knockout mice are glucose tolerant due to a compensatory increase in glucose uptake in liver (*Katz et al., 1995*; *Ranalletta et al., 2005*). We observed increased systemic glycolysis in AdHk2KO mice (*Figure 4—figure supplement 3F*). Global ablation of HK2 mice may provoke an even stronger compensatory response to maintain systemic glucose homeostasis. This may also explain the observation that global heterozygous *Hk2* knockout mice display better glucose handling than wild-type controls (*Heikkinen et al., 1999*).

We note that *Hk2* knockout did not completely phenocopy to the effect of obesity in mice and humans. While HK2 was decreased approximately 75% in vWAT of AdHk2KO mice (*Figure 4A*), HK2 expression in HFD-fed mice and HK activity in oWAT from obese patients were decreased only ~60% and ~30%, respectively (*Figure 1B and G*). Is the reduction in HK2 in obese mice and patients sufficient to contribute to systemic insulin insensitivity and glucose intolerance? A previous study showed that adipose-specific *Rab10* knockout mice display reduced GLUT4 translocation to the plasma membrane and thus a ~50% reduction in insulin-stimulated glucose disposal, in isolated adipocytes (*Vazirani et al., 2016*). Importantly, adipose-specific *Rab10* knockout mice failed to suppress hepatic glucose production and were thus insulin insensitive, indicating that a limited disruption in glucose metabolism in adipose tissue can significantly impact systemic insulin sensitivity and glucose homeostasis. Thus, the partial loss of HK2 observed in obese mice and patients may be sufficient to impact systemic insulin sensitivity and thereby glucose homeostasis.

We found an HK2-R42H mutation in naturally hyperglycemic Mexican cavefish. To date, no human monogenic disease has been reported associated with mutations in the *HK2* gene. However, 12 R42W and 4 R42Q alleles are reported in the genomAD database (*Karczewski et al., 2020*). Further studies are required to determine whether these alleles are loss-of-function and associated with diabetes.

## Materials and methods

### Mouse

Wild-type C57BL/6JRj mice were purchased from JANVIER LABS (Le Genest-Saint-lsle, France). Mice carrying *Hk2* with exons 4–10 flanked by loxP sites (*Hk2*$^{fl/fl}$) were previously described (*Patra et al., 2013*). *Adipoq-CreER*$^{T2}$ mice were provided by Prof. Stefan Offermanns (MPI-HLR, Germany)(*Sassmann et al., 2010*). *Hk2*$^{fl/fl}$ mice were crossed with *Adipoq-CreER*$^{T2}$ mice, and resulting Cre positive *Hk2*$^{fl/+}$ mice were crossed with *Hk2*$^{fl/fl}$ mice to generate adipocyte-specific *Hk2* knockout (*Adipoq-CreER*$^{T2}$ positive *Hk2*$^{fl/fl}$) mice (AdHk2KO). *Hk2* knockout was induced by *i.p. injection of* 1 mg/mouse tamoxifen (Sigma-Aldrich, St. Louis, Missouri) in corn oil (Sigma-Aldrich, St. Louis, Missouri) for 7 days. Littermate *Cre* negative animals were used as a control. Control mice were also treated with tamoxifen.

Mice were housed at 22 °C in a conventional facility with a 12 hours light/dark cycle with unlimited access to water, and normal diet (ND, KLIBA NAFAG, Kaiseraugst, Switzerland) or high-fat diet (HFD: 60% kcal % fat NAFAG 2127, KLIBA, Kaiseraugst, Switzerland). Only male mice between 6 and 17 weeks of age were used for experiments.

### Plasmids

pTwist-surface HK2 and pTwist-Molino HK2 plasmids were purchased from Twist Bioscience (South San Francisco, California). The R42H mutation was introduced into the pLenti-CMV-ratHK2 (*DeWaal et al., 2018*) by PCR using the oligos 5'atttctaggcACttccggaaggagatggagaaag3' and 5'cttccggaaG Tgcctagaaatctccagaagggtc3'. The desired sequence change was confirmed.*Figure 4—figure supplement 1*.

### Cell culture

3T3-L1 and HEK293T cells were obtained from ATCC. We perform mycoplasma contamination every three month. The cell lines used were mycoplasma negative. HEK293T cells were cultured in M1 medium composed of DMEM high glucose (Sigma-Aldrich, St. Louis, Missouri) supplemented with 4 mM glutamine (Sigma-Aldrich, St. Louis, Missouri), 1 mM sodium pyruvate (Sigma-Aldrich, St. Louis, Missouri), 1 x penicillin and streptomycin (Sigma-Aldrich, St. Louis, Missouri), and 10% FBS (ThermoFisher Scientific, Waltham, Massachusetts). 3T3-L1 cells were cultured and differentiated as previously described (*Zebisch et al., 2012*). In brief, 3T3-L1 preadipocyte cells were maintained in M1 medium at 37 °C incubator with 5% $CO_2$. For differentiation, cells were maintained in M1 medium for 2 days after reaching confluence. The cells were then transferred to M2 medium composed of M1 medium supplemented with 1.5 µg/mL insulin (Sigma-Aldrich, St. Louis, Missouri), 0.5 mM IBMX (AdipoGen LIFE SCIENCES, Liestal, Switzerland), 1 µM dexamethasone (Sigma-Aldrich, St. Louis, Missouri), and 2 µM rosiglitazone (AdipoGen LIFE SCIENCES, Liestal, Switzerland), defined as day 0 post-differentiation. After 2 days, the cells were transferred to M3 medium (M1 with 1.5 µg/mL

insulin). At day 4 post-differentiation, cells were transferred back to M2 medium. From day 6 post-differentiation, cells were maintained in M3 with medium change every 2 days.

For *Hk2* knockdown, MISSION shRNA (TRCN0000280118) or control pLKO plasmid were purchased from Sigma-Aldrich (St. Louis, Missouri) and co-transfected with psPAX2 (a gift from Didier Trono: Addgene plasmid # 12260) and pCMV-VSV-G (*Stewart et al., 2003*; a gift from Robert Weinberg: Addgene plasmid # 8454) into HEK293T cells. Supernatants containing lentivirus were collected one day after transfection, and used to infect undifferentiated 3T3-L1 cells. Transduced cells were selected by puromycin (Thermo Fisher Scientific, Waltham, Massachusetts). For all experiments, 8–14 days post-differentiated cells were used.

For HK2 overexpression in HEK293T cells, plasmids were transfected with jetPRIME (Polyplus-transfection, Illkirch-Graffenstaden, France) following manufacturer's instructions.

## Human biopsies

Omental white adipose tissue (oWAT) biopsies were obtained from lean subjects with normal fasting glucose level and body mass index (BMI) <27 kg/m$^2$, from obese non-diabetic subjects with BMI >35 kg/m$^2$ HbA1c<6.0%, and from obese diabetic sujects with BMI >35 kg/m$^2$ HbA1c>6.1% (*Supplementary file 2*). All subjects gave informed consent before the surgical procedure. Patients were fasted overnight and underwent general anesthesia. All oWAT specimens were obtained between 8:30 and 12:00 am, snap-frozen in liquid nitrogen, and stored at –80 °C for subsequent use.

## Proteomics

Tissues were pulverized and homogenized in a tissue lysis buffer containing 100 mM Tris (VWR, Radnor, Pennsylvania)-HCl (Merck, Burlington, Massachusetts) pH7.5, 2 mM EDTA (Sigma-Aldrich, St. Louis, Missouri), 2 mM EGTA (Sigma-Aldrich, St. Louis, Missouri), 150 mM NaCl (Sigma-Aldrich, St. Louis, Missouri), 1% Triton X-100 (Fluka Chemie, Buchs, Switzerland), cOmplete inhibitor cocktail (Sigma-Aldrich, St. Louis, Missouri) and PhosSTOP (Sigma-Aldrich, St. Louis, Missouri). Proteins were precipitated by trichloroacetic acid (Sigma-Aldrich, St. Louis, Missouri) and the resulting protein pellets were washed with cold acetone (Merck, Burlington, Massachusetts). 25 µg of peptides were labeled with isobaric tandem mass tags (TMT 10-plex, Thermo Fisher Scientific, Waltham, Massachusetts) as described previously (*Ahrné et al., 2016*). Shortly, peptides were resuspended in 20 µl labeling buffer (2 M urea (Sigma-Aldrich, St. Louis, Missouri), 0.2 M HEPES (Sigma-Aldrich, St. Louis, Missouri), pH 8.3) and 5 µL of each TMT reagent were added to the individual peptide samples followed by a 1 hour incubation at 25 °C. To control for ratio distortion during quantification, a peptide calibration mixture consisting of six digested standard proteins mixed in different amounts was added to each sample before TMT labeling. To quench the labelling reaction, 1.5 µL aqueous 1.5 M hydroxylamine solution (Merck, Burlington, Massachusetts) was added and samples were incubated for another 10 min at 25 °C followed by pooling of all samples. The pH of the sample pool was increased to 11.9 by adding 1 M phosphate buffer pH 12 (Sigma-Aldrich, St. Louis, Missouri) and incubated for 20 min at 25 °C to remove TMT labels linked to peptide hydroxyl groups. Subsequently, the reaction was stopped by adding 2 M hydrochloric acid (Merck, Burlington, Massachusetts) and until a pH <2 was reached. Finally, peptide samples were further acidified using 5% TFA (Thermo Fisher Scientific, Waltham, Massachusetts), desalted using Sep-Pak Vac 1 cc (50 mg) C18 cartridges (Waters, Milford, Massachusetts) according to the manufacturer's instructions and dried under vacuum.

TMT-labeled peptides were fractionated by high-pH reversed phase separation using a XBridge Peptide BEH C18 column (3,5 µm, 130 Å, 1 mm x 150 mm, Waters, Milford, Massachusetts) on an 1260 Infinity HPLC system (Agilent Technologies, Santa Clara, California). Peptides were loaded on column in buffer A (20 mM ammonium formate (Sigma-Aldrich, St. Louis, Missouri) in water, pH 10) and eluted using a two-step linear gradient from 2% to 10% in 5 min and then to 50% buffer B (20 mM ammonium formate (Sigma-Aldrich, St. Louis, Missouri) in 90% acetonitrile (Thermo Fisher Scientific, Waltham, Massachusetts), pH 10) over 55 min at a flow rate of 42 µl/min. Elution of peptides was monitored with a UV detector (215 nm, 254 nm) and a total of 36 fractions were collected, pooled into 12 fractions using a post-concatenation strategy as previously described (*Wang et al., 2011*) and dried under vacuum.

Dried peptides were resuspended in 20 µl of 0.1% aqueous formic acid (Sigma-Aldrich, St. Louis, Missouri) and subjected to LC–MS/MS analysis using a Q Exactive HF Mass Spectrometer fitted with an

EASY-nLC 1000 (both Thermo Fisher Scientific, Waltham, Massachusetts) and a custom-made column heater set to 60 °C. Peptides were resolved using a RP-HPLC column (75 μm×30 cm) packed in-house with C18 resin (ReproSil-Pur C18–AQ, 1.9 μm resin; Dr. Maisch GmbH, Ammerbuch, Germany) at a flow rate of 0.2 μL/min. The following gradient was used for peptide separation: from 5% B to 15% B over 10 min to 30% B over 60 min to 45% B over 20 min to 95% B over 2 min followed by 18 min at 95% B. Buffer A was 0.1% formic acid (Sigma-Aldrich, St. Louis, Missouri) in water and buffer B was 80% acetonitrile (Thermo Fisher Scientific, Waltham, Massachusetts), 0.1% formic acid (Sigma-Aldrich, St. Louis, Missouri) in water.

The mass spectrometer was operated in DDA mode with a total cycle time of approximately 1 s. Each MS1 scan was followed by high-collision-dissociation (HCD) of the 10 most abundant precursor ions with dynamic exclusion set to 30 s. For MS1, 3e6 ions were accumulated in the Orbitrap over a maximum time of 100 ms and scanned at a resolution of 120,000 FWHM (at 200 m/z). MS2 scans were acquired at a target setting of 1e5 ions, accumulation time of 100 ms and a resolution of 30,000 FWHM (at 200 m/z). Singly charged ions and ions with unassigned charge state were excluded from triggering MS2 events. The normalized collision energy was set to 35%, the mass isolation window was set to 1.1 m/z and one microscan was acquired for each spectrum.

The acquired raw-files were converted to the mascot generic file (mgf) format using the msconvert tool (part of ProteoWizard, version 3.0.4624 (2013-6-3)). Using the MASCOT algorithm (Version 2.4.1, Matrix Science, Boston Massachusetts), the mgf files were searched against a decoy database containing normal and reverse sequences of the predicted SwissProt entries of *Mus musculus* (https://www.ebi.ac.uk/, release date 2014/11/24), the six calibration mix proteins (*Ahrné et al., 2016*) and commonly observed contaminants (in total 50214 sequences for *Mus musculus*) generated using the SequenceReverser tool from the MaxQuant software (Version 1.0.13.13). The precursor ion tolerance was set to 10 ppm and fragment ion tolerance was set to 0.02 Da. The search criteria were set as follows: full tryptic specificity was required (cleavage after lysine or arginine residues unless followed by proline), 3 missed cleavages were allowed, carbamidomethylation (C) and TMT6plex (K and peptide N-terminus) were set as fixed modification and oxidation (M) as a variable modification. Next, the database search results were imported into the Scaffold Q+software (version 4.3.2, Proteome Software Inc, Portland, Oregon) and the protein false identification rate was set to 1% based on the number of decoy hits. Proteins that contained similar peptides and could not be differentiated based on MS/MS analysis alone were grouped to satisfy the principles of parsimony. Proteins sharing significant peptide evidence were grouped into clusters. Acquired reporter ion intensities in the experiments were employed for automated quantification and statically analysis using a modified version of our in-house developed SafeQuant R script (*Ahrné et al., 2016*). This analysis included adjustment of reporter ion intensities, global data normalization by equalizing the total reporter ion intensity across all channels, summation of reporter ion intensities per protein and channel, calculation of protein abundance ratios and testing for differential abundance using empirical Bayes moderated t-statistics. Finally, the calculated p-values were corrected for multiple testing using the Benjamini−Hochberg method (q-value). Deregulated proteins were selected by log2(fold change)>0.6 or log2(-fold change)<–0.6, q-value <0.01.

## Immunoblots

Tissues or cells were homogenized in a lysis buffer containing 100 mM Tris (Merck, Burlington, Massachusetts) pH7.5, 2 mM EDTA (Sigma-Aldrich, St. Louis, Missouri), 2 mM EGTA (Sigma-Aldrich, St. Louis, Missouri), 150 mM NaCl (Sigma-Aldrich, St. Louis, Missouri), 1% Triton X-100 (Fluka Chemie, Buchs, Switzerland), cOmplete inhibitor cocktail (Sigma-Aldrich, St. Louis, Missouri) and PhosSTOP (Sigma-Aldrich, St. Louis, Missouri). Protein concentration was determined by the Bradford assay (Bio-rad), and equal amounts of protein were separated by SDS-PAGE, and transferred onto nitrocellulose membranes (GE Healthcare, Chicago, Illinois). Antibodies used in this study were as follows: AKT (Cat#4685 or Cat#2920), AKT-pS473 (Cat#4060), AKT-pT308 (Cat#13038), PRAS40-pT246 (Cat#2997), PRAS40 (Cat#2691), HK2 (Cat#2867), S6-pS240/244 (Cat#5364), S6 (Cat#2217), FASN (Cat#3189), ACC (Cat#3662), HA tag (Cat#3724) and ChREBP (Cat#58069) from Cell Signaling Technology (Danvers, Massachusetts), GLUT4 (Cat# NBP2-22214, Bio-Techne, Abingdon, United Kingdom), HSP90, (Cat#sc-13119, Santa Cruz Biotechnology, Dallas, Texas), CALNEXIN (Cat#ADI-SPA-860-F, Enzo Life Sciences, Farmingdale, New York) and ACTIN

(Cat#MAB1501, Merck, Burlington, Massachusetts). For quantification, the specific signals were normalized to a loading control.

## Metabolomics

Tissues (25–30 mg) were finely ground in a cryogenic grinding before the extraction. Metabolite extraction was performed, in a mixture ice/dry ice, by a cold two-phase methanol–water–chloroform extraction (*Elia et al., 2017*; *van Gorsel et al., 2019*). The samples were resuspended in 900 µl of precooled methanol/water (5/3) (v/v) and 100 µL of $^{13}$C yeast internal standard. Afterwards, 500 µl of precooled chloroform was added to each sample. Samples were vortexed for 10 min at 4 °C and then centrifuged (max. speed, 10 min, 4 °C). The methanol–water phase containing polar metabolites was separated and dried using a vacuum concentrator at 4 °C overnight and stored at −80 °C.

For the detection of the pentose phosphate pathway and glycolysis intermediates by LC-MS, a 1290 Infinity II liquid chromatography (Agilent Technologies, Santa Clara, California) with a thermal autosampler set at 4 °C, coupled to a Q-TOF 6546 mass spectrometer (Agilent Technologies, Santa Clara, California) was used. Samples were resuspended in 100 µL of 50% methanol and a volume of 5 µL and 20 µL of sample were injected on Agilent InfinityLab Poroshell 120 HILIC-Z column, 2.1 mm × 150 mm, 2.7 µm, PEEK-lined. The separation of metabolites was achieved at 50 °C with a flow rate of 0.25 ml/min. A gradient was applied for 32 min (solvent A: 10 mM ammonium acetate in water with 2.5 µM InfinityLab Deactivator Additive, pH = 9 – solvent B: 10 mM ammonium acetate in water/acetonitrile 15:85 (v:v) with 2.5 µM InfinityLab Deactivator Additive, pH = 9) to separate the targeted metabolites (0 min: 96%B, 2 min: 96%B, 5.5 min: 88%B, 8.5 min: 88%B, 9 min: 86%B, 14 min: 86%B, 19 min: 82%B; 25 min: 65%B, 27 min: 65%B, 28 min: 96%B; 32 min: 96%B).

The MS operated in negative full scan mode (m/z range: 50–1200) using a shealth gas temperature of 350 °C (12 L/min) and a gas temperature at 225 °C (13 L/min). The nebulizer was set at 35 psi, the fragmentor at 125 V and the capillary at 3500 V. Data was collected using the Masshunter software (Agilent Technologies, Santa Clara, California) and normalized by $^{13}$C yeast internal standard and the protein content.

For the detection of Acetyl-CoA by LC-MS, a Dionex UltiMate 3000 LC System (Thermo Fisher Scientific, Waltham, Massachusetts) with a thermal autosampler set at 4 °C, coupled to a Q Exactive Orbitrap mass spectrometer (Thermo Fisher Scientific, Waltham, Massachusetts) was used. Samples were resuspended in 100 µL of 50% MeOH and a volume of 10 µl of sample was injected on a C18 column (Acquity UPLC HSS T3 1.8 µm 2.1x100 mm). The separation of metabolites was achieved at 40 °C with a flow rate of 0.25 ml/min. A gradient was applied for 40 min (solvent A: 10 mM Tributyl-Amine, 15 mM acetic acid – solvent B: Methanol) to separate the targeted metabolites (0 min: 0%B, 2 min: 0%B, 7 min: 37%B, 14 min: 41%B, 26 min: 100%B, 30 min: 100%B, 31 min: 0%B; 40 min: 0%B).

The MS operated in negative full scan mode (m/z range: 70–1050 and 750–850 from 5 to 25 min) using a spray voltage of 4.9 kV, capillary temperature of 320 °C, sheath gas at 50.0, auxiliary gas at 10.0. Data was collected using the Xcalibur software (Thermo Fisher Scientific, Waltham, Massachusetts) and analyzed with Matlab for the correction of natural abundance. Data were normalized by $^{13}$C yeast internal standard and the protein content.

## RNA isolation and quantitative RT-PCR

Total RNA was isolated with TRIzol reagent (Sigma-Aldrich, St. Louis, Missouri) and RNeasy kit (Qiagen, Hilden, Germany). RNA was reverse-transcribed to cDNA using iScript cDNA synthesis kit (Bio-Rad Laboratories, Hercules, California). Semiquantitative real-time PCR analysis was performed using fast SYBR green (Applied Biosystems, Waltham, Massachusetts). Relative expression levels were determined by normalizing each CT values to *Tbp* using the ΔΔCT method. The sequence for the primers (Microsynth, Balgach, Switzerland) used in this study was as follows:

> *Fasn*-fw: 5′GCTGCGGAAACTTCAGGAAAT3′,
> *Fasn*-rv: 5′AGAGACGTGTCACTCCTGGACTT3′,
> *Acc*-fw: 5′AAGGCTATGTGAAGGATG3′,
> *Acc*-rv: 5′CTGTCTGAAGAGGTTAGG3′,
> *G6pc*-fw: 5′CCATGCAAAGGACTAGGAACAA3′,
> *G6pc*-rv: 5′TACCAGGGCCGATGTCAAC3′,
> *Pepck*-fw: 5′CCACAGCTGGTGCAGAACA3′,

*Pepck*-rv: 5'GAAGGGTCGATGGCAAA3',
*Chrebp*-fw: 5'CACTCAGGGAATACACGCCTAC3',
*Chrebp*-rv: 5'ATCTTGGTCTTAGGGTCTTCAGG3',
*Chrebp-alpha*-fw: 5'CGACACTCACCCACCTCTTC3',
*Chrebp-alpha*-rv: 5'TTGTTCAGCCGGATCTTGTC3',
*Chrebp-beta*-fw: 5'TCTGCAGATCGCGTGGAG3',
*Chrebp-beta*-rv: 5'CTTGTCCCGGCATAGCAAC3',
*Hk2*-fw: 5'ACGGAGCTCAACCAAAACCA3',
*Hk2*-rv: 5'TCCGGAACCGCCTAGAAATC3',
*Foxk1*-fw: 5'GGCTGTCACTCAGAATGGAA3',
*Foxk1*-rv: 5'GAGGCAGATGTGGTAGTGGAG3',
*Foxk2*-fw: 5'CCACGGGAACTATCAGTGCT3',
*Foxk2*-rv: 5'GTCATCCTTTGGGCTGTCTC3',
*Lep*-fw: 5'TCACACACGCAGTCGGTATC3',
*Lep*-rv: 5'ACTCAGAATGGGGTGAAGCC3',
*Adipoq*-fw: 5'TGACGACACCAAAAGGGCTC3',
*Adipoq*-rv: 5'ACGTCATCTTCGGCATGACT3',
*aP2*-fw: 5'TCGGTTCCTGAGGATACAAGAT3',
*aP2*-rv: 5'TTTGATGACTGTGGGATTGAAG3',
*Ucp1*-fw: 5'TGATGAAGTCCAGACAGACAGTG3',
*Ucp1*-rv: 5'TTATTCGTGGTCTCCCAGCATAG3',
*Dio2*-fw: 5'GAGGAAGGAAGAAGAGGAAGCAA3',
*Dio2*-rv: 5'TTCTTCCAGTGTTTTGGACATGC3',
*Tbp*-fw: 5'TGCTGTTGGTGATTGT3',
*Tbp*-rv: 5'CTTGTGTGGGAAAGAT3'.

## Hexokinase assay

For measuring hexokinase activity of surface fish HK2, Molino HK2, wild-type rat HK2 or rat HK2-R42H, HEK293T cells were transfected with plasmids containing these HK2 sequences, and cells were harvested at 24 hours after transfection. Hexokinase activity were measured with cell lysates or tissue lysates with a hexokinase assay kit (Abcam, Cambridge, United Kingdom) following manufacturer's instructions. The final glucose concentration in the hexokinase assay was 4 mM. The hexokinase activities were normalized to the protein content in lysates.

## Seahorse analyses

Measurements were performed with an XF96 Extracellular Flux Analyzer (Seahorse Bioscience of Agilent Technologies, Santa Clara, California) following manufacturer's instructions.

## Lactate measurement

Differentiated adipocytes were starved serum overnight and stimulated with 100 nM insulin for 2 hours. Extracellular lactate was measured with the Lactate Pro 2 analyzer (Axon Lab, Stuttgart, Germany).

## $^3$H-2DG uptake assay in vitro

Differentiated adipocytes were starved for serum for 5 hours and then incubated in Krebs Ringer Phosphate Hepes (KRPH) buffer composed of 0.6 mM $Na_2HPO_4$ (Fluka Chemie, Buchs, Switzerland), 0.4 mM $NaH_2PO_4$ (Fluka Chemie, Buchs, Switzerland), 120 mM NaCl (Sigma-Aldrich, St. Louis, Missouri), 6 mM KCl (Merck), 1 mM $CaCl_2$ (Merck, Burlington, Massachusetts), 1.2 mM $MgSO_4$ (Merck, Burlington, Massachusetts), 12.5 mM HEPES (Thermo Fisher Scientific, Waltham, Massachusetts), 0.2% fatty acid-free BSA (Sigma-Aldrich, St. Louis, Missouri) pH7.4 with or without 100 nM insulin (Sigma-Aldrich, St. Louis, Missouri) for 20 min. Cells were incubated with cold 50 µM 2DG (Sigma-Aldrich, St. Louis, Missouri) containing 0.25 µCi $^3$H-2-deoxyglucose (2DG, Perkin Elmer, Waltham, Massachusetts) for 5 min and washed three times with cold PBS (Sigma-Aldrich, St. Louis, Missouri). Cells were lysed in the tissue lysis buffer and cleared by centrifugation at 14,000 g for 10 min. Incorporated $^3$H-2DG was measured with a scintillation counter (Perkin Elmer, Waltham, Massachusetts).

### Insulin tolerance test, glucose tolerance test, pyruvate tolerance test

For the insulin and glucose tolerance tests, mice were fasted for 6 hours and insulin Humalog (i.p. 0.75 or 0.5 U/kg body weight, Lilly, Indianapolis, Indiana) or glucose (2 g/kg body weight, Sigma-Aldrich, St. Louis, Missouri) was given, respectively. For the pyruvate tolerance test, mice were fasted for 15 hours and pyruvate (2 g/kg body weight, Sigma-Aldrich, St. Louis, Missouri) was administered. Blood glucose was measured with a blood glucose meter (Accu-Check, Roche Diabetes Care, Indiana polis, Indiana).

### Hyperinsulinemic-euglycemic clamp

Hyperinsulinemic-euglycemic clamp was performed as previously described (*Smith et al., 2018*). In brief, mice were fasted for 6 hours and anesthetized by i.p. injection of 6.25 mg/kg acetylpromazine (Fatro, Bologna, Italy), 6.25 mg/kg midazolam (Sintetica, Val-de-Travers, Switzerland) and 0.31 mg/kg fentanyl (Mepha, Aesch, Switzerland). An infusion needle were placed into the tail vein and $^3$H-glucose (Parkin Elmer, Waltham, Massachusetts) was infused for 60 min to achieve steady-state levels. Thereafter, the hyperinsulinemic clamp started with a bolus dose (3.3mU) and a constant infusion of insulin (0.09 mU/min) and $^3$H-glucose. A variable infusion of 12.5% D-glucose (Sigma-Aldrich, St. Louis, Missouri) was used to maintain euglycemia. Blood glucose was measured with a a blood glucose meter (Accu-Check, Roche Diabetes Care, Indiana polis, Indiana) every 5–10 min and the glucose infusion rate was adjusted to maintain euglycemia. Blood samples were taken to determine steady-state levels of [$^3$H]-glucose. After 90 min from the start of the insulin clamp, $^{14}$C-2-Deoxyglucose (Parkin Elmer, Waltham, Massachusetts) was i.p. administered to assess tissue-specific glucose uptake. Mice were euthanized by cervical dislocation and the organs were removed and frozen. $^3$H-glucose and $^{14}$C-2-DG phosphate counts in plasma and tissues were measured by a scintillation counter.

### Hematoxylin and eosin staining

Tissues were fixed in 4% formalin (Leica Biosystems, Wetzlar, Germany), embedded in paraffin (Leica Biosystems, Wetzlar, German), and sliced into 3-µm-thick section. Tissue sections were stained with Hematoxylin (Sigma-Aldrich, St. Louis, Missouri) and eosin (Waldeck, Münster, Germany), and imaged by Axio Scan. Z1 slidescanner (Zeiss, Oberkochen, Germany).

### In vivo lipogenesis

Ad libitum-fed mice were i.p. injected with 1 mCi $^3$H-H$_2$O (American Radiolabeled Chemicals, St. Louis, Missouri) and sacrificed with i.p.160 mg/kg ketamine (Streuli Pharma, Uznach, Switzerland) and 24 mg/kg xylazine (Streuli Pharma, Uznach, Switzerland). For triglyceride extraction, plasma samples were mixed with 1 mL of 2-propanol (Merck, Burlington, Massachusetts) /n-hexane (Merck, Burlington, Massachusetts) /1 N H$_2$SO$_4$ (Sigma-Aldrich, St. Louis, Missouri) (4:1:1) and incubated for 30 min. ddH2O and n-hexane were added and the resulting n-hexane phase was collected, dried, and counted by a scintillation counter.

### Ex vivo lipogenesis

vWAT explants were washed and incubated with low glucose DMEM (Sigma-Aldrich, St. Louis, Missouri) supplemented with 2% fatty acid-free BSA (Sigma-Aldrich, St. Louis, Missouri) and 20 mM HEPES (Thermo Fisher Scientific, Waltham, Massachusetts). Explants were further washed with a buffer containing 10 mM HEPES (Thermo Fisher Scientific, Waltham, Massachusetts), 116 mM NaCl (Sigma-Aldrich, St. Louis, Missouri), 4 mM KCl (Merck, Burlington, Massachusetts), 1.8 mM CaCl$_2$ (Merck, Burlington, Massachusetts), 1 mM MgCl$_2$ (Fluka Chemie, Buchs, Switzerland), 4.5 mM D-glucose (Sigma-Aldrich, St. Louis, Missouri), and 2.5% fatty acid-free BSA (Sigma-Aldrich, St. Louis, Missouri). 2 µCi of $^3$H-H$_2$O (Perkin Elmer) was added in the absence or presence of 100 nM insulin (Sigma-Aldrich, St. Louis, Missouri). After 4.5 hours, explants were washed with cold PBS (Sigma-Aldrich, St. Louis, Missouri) three times and frozen in liquid nitrogen. Triglycerides were extracted as described above and 1/3 of the n-hexane phase was used for triglyceride measurement. For fatty acid extraction, the remaining hexane phase was deacylated with ethanol (Merck, Burlington, Massachusetts):water:-saturated KOH (Merck, Burlington, Massachusetts) (20:1:1) at 80 °C for 1 hour, neutralized with formic acid (Sigma-Aldrich, St. Louis, Missouri). Fatty acid was extracted with n-hexane and dried. The incorporation of $^3$H was counted by a scintillation counter and normalized to tissue weight.

## Ex vivo NEFA release

vWAT explants were washed and incubated with low glucose DMEM (Sigma-Aldrich, St. Louis, Missouri) supplemented with 2% fatty acid-free BSA (Sigma-Aldrich, St. Louis, Missouri) and 20 mM HEPES (Thermo Fisher Scientific, Waltham, Massachusetts) for 2 hours. Explants were washed twice with Krebs-Ringer phosphate buffer (0.6 mM $Na_2HPO_4$ (Fluka Chemie, Buchs, Switzerland), 0.4 mM $NaH_2PO_4$ (Merck, Burlington, Massachusetts), 120 mM NaCl (Sigma-Aldrich, St. Louis, Missouri), 6 mM KCl (Merck, Burlington, Massachusetts), 1 mM $CaCl_2$ (Merck, Burlington, Massachusetts), 1.2 mM $MgSO_4$ (Merck, Burlington, Massachusetts), 70 mM HEPES (Thermo Fisher Scientific, Waltham, Massachusetts), 5 mM glucose (Sigma-Aldrich, St. Louis, Missouri), 2% fatty acid-free BSA (Sigma-Aldrich, St. Louis, Missouri), pH7,4). Explants were treated with DMSO or 10 µM isoproterenol (Sigma-Aldrich, St. Louis, Missouri) in the absence or presence of 100 nM insulin (Sigma-Aldrich, St. Louis, Missouri) for 2 hours. Explants were transferred to chloroform (Sigma-Aldrich, St. Louis, Missouri): methanol (Sigma-Aldrich, St. Louis, Missouri):100% acetic acid (Merck, Burlington, Massachusetts) (200:100:3) and incubated at 37 °C for 1 hour, and protein concentration was determined by BCA assay (Pierce). Non-esterified fatty acid levels in conditioned media were determined by a colorimetric assay kit (Sigma-Aldrich, St. Louis, Missouri), and normalized with protein amounts in the explants.

## AHA-incorporation

vWAT from ND- or HFD-fed mice were immediately incubated in low glucose DMEM (Sigma-Aldrich, St. Louis, Missouri) containing 50 µM azidohomoalanine (AHA, Thermo Fisher Scientific, Waltham, Massachusetts) for 30 min. The tissues were frozen, pulverized, lysed by bio-rupture and clarified by centrifugation at 15,000 g for 15 min twice. 100 µg of protein was used for CLICK reaction (Thermo Fisher Scientific, Waltham, Massachusetts) chemistry following the manufacturer's instructions. Biotinylated protein was pulled-down after mixing with streptavidin magnetic beads (Thermo Fisher Scientific, Waltham, Massachusetts) for 2 hours at 4 °C. The pulled-down protein was digested with trypsin (Promega, Madison, Wisconsin). The digested peptides were acidified using 5% TFA (Thermo Fisher Scientific, Waltham, Massachusetts) and desalted using C18 columns (Waters, Milford, Massachusetts). The eluted peptides were dried and analyzed by mass spectrometry.

## Metabolites measurement

Plasma insulin levels were measured by ultrasensitive mouse insulin ELISA kit (Crystal Chem, Downers Grove, Illinois) according to the manufacturer's instructions. Plasma Leptin levels were measured by mouse Leptin ELISA kit (Crystal Chem, Downers Grove, Illinois) according to the manufacturer's instructions. Plasma Adiponectin levels were measured by mouse Adiponectin ELISA kit (Crystal Chem, Downers Grove, Illinois) according to the manufacturer's instructions. Hepatic triglyceride levels were measured using a triglyceride assay kit (Abcam, Cambridge, United Kingdom) according to the manufacturer's instructions. Plasma triglyceride and cholesterol levels were measured by a biochemical analyzer (Cobas c III analyser, Roche Diagnostics, Indianapolis, Indiana). Plasma NEFA levels were measured by colorimetric NEFA (Sigma-Aldrich, St. Louis, Missouri).

## Body composition measurement

Body composition was measured by nuclear magnetic resonance imaging (Echo Medical Systems, Houston, Texas).

## Study approval

All animal experiments were performed in accordance with federal guidelines for animal experimentation and were approved by the Kantonales Veterinäramt of the Kanton Basel-Stadt (#31986–3040) or KU Leuven animal ethical committee (#206/2020). For human biopsies, the study protocol was approved by the Ethikkommission Nordwest- und Zentralschweiz (EKNZ, BASEC 2016–01040).

## Statistics

Sample size was chosen according to our previous studies and published reports in which similar experimental procedures were described. The investigators were not blinded to the treatment groups except for the hyperinsulinemic-euglycemic clamp study. All data are shown as the mean ± SEM. Sample numbers are indicated in each figure legend. For mouse experiments, *n* represents the

number of animals, and for cell culture experiments, N indicates the number of independent experiments. To determine the statistical significance between 2 groups, an unpaired two-tailed Student's t test, Mann-Whitney test, or multiple t test was performed. For more than three groups, one-way ANOVA was performed. For ITT, GTT, PTT, glucose infusion rate, weigh curve data, two-way ANOVA was performed. All statistical analyses were performed using GraphPad Prism 9 (GraphPad Software, San Diego, California). A p value of less than 0.05 was considered statistically significant.

## Acknowledgements

We thank Stefan Offermanns (MPI-HLR, Germany), Didier Trono (EPFL, Switzerland), Robert Weinberg (MIT, USA), Christine Riggenbach (St. Claraspital), Christoph Handschin (Biozentrum), the Imaging Core Facility (Biozentrum), and the Proteomics Core Facility (Biozentrum) for providing reagents and technical support. We acknowledge support from the Swiss National Science Foundation (project 179569 and NCCR 182880 to MNH and 161510 to MS), KU Leuven internal fund (Project STG/20/020 and C14/22/116 to MS), European Foundation of the study of Diabetes/Novo Nordisk Foundation (MS), a FWO PhD fellowship (AV), FWO (Project G098120N to SMF), Fonds Baillet Latour (SMF), The Louis Jeantet Foundation (MNH), and the Canton of Basel (MNH). None of the funding sources was involved in study design, data collection and interpretation, or the decision to submit the work for publication.

## Additional information

### Competing interests

Bettina K Wölnerhanssen, Anne Christin Meyer-Gerspach: is affiliated with St. Clara Research Ltd. The author has no other competing interests to declare. The other authors declare that no competing interests exist.

### Funding

| Funder | Grant reference number | Author |
|---|---|---|
| Schweizerischer Nationalfonds zur Förderung der Wissenschaftlichen Forschung | 179569 | Michael N Hall |
| Schweizerischer Nationalfonds zur Förderung der Wissenschaftlichen Forschung | NCCR182880 | Michael N Hall |
| Schweizerischer Nationalfonds zur Förderung der Wissenschaftlichen Forschung | 161510 | Mitsugu Shimobayashi |
| KU Leuven | STG/20/020 | Mitsugu Shimobayashi |
| KU Leuven | C14/22/116 | Mitsugu Shimobayashi |
| European Foundation for the Study of Diabetes | | Mitsugu Shimobayashi |
| Fonds Wetenschappelijk Onderzoek | | Anke Vandekeere |
| Fonds Wetenschappelijk Onderzoek | G098120N | Sarah-Maria Fendt |
| Fonds Baillet Latour | | Sarah-Maria Fendt |
| Louis-Jeantet Foundation | | Michael N Hall |

| Funder | Grant reference number | Author |
|--------|------------------------|--------|
| Universität Basel | | Michael N Hall |

The funders had no role in study design, data collection and interpretation, or the decision to submit the work for publication.

## Author contributions

Mitsugu Shimobayashi, Conceptualization, Resources, Data curation, Formal analysis, Supervision, Funding acquisition, Validation, Investigation, Methodology, Writing – original draft, Project administration, Writing – review and editing; Amandine Thomas, Sunil Shetty, Data curation, Formal analysis, Investigation, Methodology; Irina C Frei, Data curation, Validation, Investigation, Methodology; Bettina K Wölnerhanssen, Resources, Data curation, Project administration; Diana Weissenberger, Danilo Ritz, Data curation, Formal analysis, Validation, Investigation, Methodology; Anke Vandekeere, Data curation, Funding acquisition, Investigation, Methodology; Mélanie Planque, Data curation, Investigation, Methodology; Nikolaus Dietz, Timm Maier, Data curation, Investigation; Anne Christin Meyer-Gerspach, Ralph Peterli, Resources, Data curation; Nissim Hay, Resources; Sarah-Maria Fendt, Data curation, Formal analysis, Funding acquisition, Project administration; Nicolas Rohner, Data curation, Formal analysis, Investigation, Methodology, Project administration; Michael N Hall, Conceptualization, Resources, Data curation, Supervision, Funding acquisition, Writing – original draft, Project administration, Writing – review and editing

## Author ORCIDs

Mitsugu Shimobayashi (iD) http://orcid.org/0000-0002-6936-0990
Anke Vandekeere (iD) http://orcid.org/0000-0001-6836-3834
Mélanie Planque (iD) http://orcid.org/0000-0001-7052-7084
Nikolaus Dietz (iD) http://orcid.org/0000-0001-7300-4342
Timm Maier (iD) http://orcid.org/0000-0002-7459-1363
Nissim Hay (iD) http://orcid.org/0000-0002-6245-3000
Sarah-Maria Fendt (iD) http://orcid.org/0000-0001-6018-9296
Nicolas Rohner (iD) http://orcid.org/0000-0003-3248-2772

## Ethics

Human subjects: Informed onset was obtained from all participants, and the study protocol was approved by the Ethikkomission Nordwest- und Zentralschweiz (EKNZ BASEC 2016-01040).
All animal experiments were performed in accordance with federal guidelines for animal experimentation and were approved by the Kantonales Veterinäramt of the Kanton Basel-Stadt (#31986 - 3040) or KU Leuven animal ethical committee (#206/2020).

## Decision letter and Author response

Decision letter https://doi.org/10.7554/eLife.85103.sa1
Author response https://doi.org/10.7554/eLife.85103.sa2

# Additional files

## Supplementary files
• MDAR checklist
• Supplementary file 1. vWAT proteomics data from 4w HFD-fed mice, compared to ND-fed mice.
• Supplementary file 2. Patients' data.
• Supplementary file 3. AHA data and RNAseq data in vWAT from 4w HFD-fed mice, compared to ND-fed mice.

## Data availability
All gel images and numerical data are uploaded as source data.

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
