## [Editor Report]

This study reveals that expression of the glycolytic enzyme hexokinase 2 (HK2) in adipocytes is decreased in obesity and is associated with glucose intolerance and insulin resistance. The authors then show that adipose selective depletion of HK2 in mice causes systemic glucose intolerance, suggesting that the decreased HK2 may contribute to metabolic dysfunction in obese humans. These studies point to a potentially important new pathway that contributes to the regulation of metabolic health.

---

## [Decision Letter]

**Decision letter after peer review:**

[Editors’ note: the authors submitted for reconsideration following the decision after peer review. What follows is the decision letter after the first round of review.]

Thank you for submitting the paper "Diet-induced loss of adipose Hexokinase 2 triggers hyperglycemia" for consideration by *eLife*. Your article has been reviewed by 3 peer reviewers, one of whom is a member of our Board of Reviewing Editors, and the evaluation has been overseen by a Senior Editor. The following individuals involved in review of your submission have agreed to reveal their identity: David E James (Reviewer #3).

Comments to the Authors:

We are sorry to say that, after consultation with the reviewers, we have decided that this work will not be considered further for publication by *eLife*.

Specifically, a key finding in your study--that adipose HK2 KO causes systemic glucose intolerance--is expected based on previous data on GLUT4, but it doesn't mimic conditions observed in obesity where the inhibition on HK2 expression is not complete. Experiments that would be required to test the effects of adipose HK2 depletion that mirrors what is observed in obesity will be difficult but may be possible. The reviews provided below address other important concerns that reflect what is viewed as a preliminary stage of the study. If you are able in the future to address the several specific points raised in the reviews with additional experiments in a detailed and compelling way, and still wish to have it evaluated by *eLife*, we will do so.

*Reviewer #1 (Recommendations for the authors):*

This is an interesting manuscript which reports effects of adipose HK2 KO in an attempt to determine the role of hexokinase in adipocytes as it relates to whole body metabolism. The data indicate that VAT HK2 is strongly downregulated in obesity/HFD, while SAT shows some decrease as well, although less of an effect than seen in VAT. The adipose tissue selective KO causes marked changes in systemic glucose tolerance. The authors conclude that this ability of adipose HK2 to modify glucose tolerance is due to its requirement for the esterification of FA that is released in lipolysis. Therefore, in HK2KO adipocytes, more FA is released into the circulation and is hypothetically able to induce gluconeogenesis by known mechanisms. The following two most important concerns are noted:

1. It is not surprising that HK2 KO in adipocytes causes glucose intolerance, as previous work has clearly demonstrated that inhibiting glucose metabolism in adipocytes has this consequence. In this regard, adipocyte GLUT4KO shows a similar phenotype and this has been studied in great detail in the past. Similar to GLUT4, the authors show here that HK2 is also decreased in obesity, and therefore could contribute to decreasing adipocyte glucose metabolism and the effect on glucose tolerance. However, a full KO of HK2 does not mimic the physiological system in obesity since HK2 is diminished but not totally inhibited in the obese state in humans and in the SAT of mice, while the KO completely eliminates the enzyme. Since glucose transport is considered to be rate limiting for glucose uptake, the argument hinges on whether the level of decrease in HK2 in obesity is actually contributing much to a decrease in glucose metabolism compared to the contribution of the decreased GLUT4. A much more informative experiment would be to study het mice where HK2 might be decreased to a more physiological level that is actually seen in obesity rather than the full KO where the result is difficult to interpret.

2. The text goes beyond the data in attributing correlation to causation. Examples are Figure 5 and Figure 6 legends that ascribe defined mechanisms to what are essentially correlations that are observed. It is not shown definitively in the data of Figure 5, for example, that de novo lipogenesis deficiency causes increased hepatic gluconeogenesis. As a matter of fact, hepatic gluconeogenesis is not measured in this Figure. And the NEFA data are correlative with the decreased adipose tissue de novo lipogenesis. Similarly, Figure 6 doesn't show mechanism, only correlation.

*Reviewer #2 (Recommendations for the authors):*

It is well established that feeding mice with western diets causes insulin resistance in liver, adipose tissue and muscle. There are many studies that propose various mechanisms for this effect. This paper describes a completely new and exciting mechanism involving changes in the expression of hexokinase 2, an important glycolytic enzyme, in adipose tissue. The authors provide sufficient evidence to show that the expression of hexokinase 2 in adipose tissue is able to regulate whole body metabolism. However, there are some issues that the authors could address prior to further consideration of the study. For example, further information that might address the mechanism for the fall in hexokinase 2 would be valuable and an explanation for the disparity between some of the measures of insulin sensitivity are also necessary.

This is a terrific study that brings new light on this heavily studied problem of metabolic dysfunction in the high fat fed mouse. The data is for the most part of very high quality and the manuscript is well written and researched.

1. The biggest issue for me with this paper is the huge discrepancy between the 'clamp' data and the other measures of insulin action. I have personally never seen such a massive defect in insulin action by clamp as this. It essentially shows that there is zero response to insulin whatsoever in these mice. Yet all other tests (GTT, ITT etc) show a relatively mild phenotype. The tissue specific glucose uptake is also not in step with this defect in that the liver defect is mild, there is no apparent defect in muscle and the only real defect is in adipose tissue, which theoretically accounts for a small proportion of whole body glucose utilisation. The other issue is that in absolute terms 2DG uptake in muscle is on par with WAT which seems worthy of questioning. I was unable to find the insulin levels generated during the clamp so I am not sure if that is a problem. I strongly urge the authors to go back over these data to see if they can spot a calculation error.

2. Given that the authors have deep proteomics data from these mice it would be interesting to run a co-regulation analysis (e.g. WGCNA) to determine if there are other proteins in the data that behave like HK2, as this might provide some mechanistic hints.

3. The mechanism for the loss of insulin regulation of lipolysis in the HK Mice is interesting. I was not particularly convinced by the glycerol data as measures of plasma glycerol are not particularly informative as it probably has a very high flux. So as a minimum I think this part needs expansion.

4. In your AHA experiments which are very nice it would be valuable if you could show some 'control' proteins that do not behave like HK2.

5. It would be fascinating to know if deletion of HK2 has affected GLUT4 translocation? I know this might be a bit of a stretch for the authors but it might be easily achievable in your 3T3-L1 cells if you could express t

*Reviewer #3 (Recommendations for the authors):*

In this manuscript, Shimobayashi et al. investigate the molecular pathogenesis of high-fat diet induced insulin resistance with a focus on adipose tissue. The authors show that short-term (4 weeks) high-fat diet is sufficient to cause glucose intolerance in male mice. Through a proteomic screen, they identify downregulation of HK2 in visceral fat as one of many changes in the adipose proteome in this timeframe. The authors observe that HK2 expression and HK activity is reduced in adipose tissue in human obesity. The authors propose that down-regulation of adipose HK2 is an early and critical step in the pathogenesis of insulin resistance. The authors also observe that the hyperglycemic Molino Cavefish sub-species carries an inactivating mutation in HK2 and propose that this contributes to hyperglycemia in this model organism. The authors generate adipose specific HK2 knockout mice and show that this is sufficient to cause peripheral insulin resistance with reduced glucose uptake in WAT and BAT as well as hepatic insulin resistance with increased glucose production. This is a well-written manuscript conducted with sophistication and technical expertise. The comprehensive phenotyping of the adipose HK2 KO mice is a strength and the phenotypes are clear and robust. However, as detailed below, previously published data not discussed in this manuscript indicates that substantial reductions in HK2 activity in vivo are insufficient to cause systemic insulin resistance and this undermines the author's conclusion that downregulation of HK2 is the critical step in high-fat diet induced changes in adipose function.

Concerns:

1. Prior literature (PMID: 10428828) indicates that global HK2 heterozygous mice demonstrated a 50% reduction in tissue HK2 activity (in both adipose and muscle) but have normal glucose homeostasis including normal glucose tolerance and insulin action. Although the authors show that complete ablation of adipose HK2 can impair systemic glucose metabolism, the previously published work calls in to question the physiological importance intermediate reductions in adipose HK2 activity that are commonly observed with obesity or dietary interventions. This is particularly true for humans where mean adipose HK activity is reduced by ~ 50% in people with obesity with substantial variance and overlap with those who are lean. The current work must be discussed with respect to this prior literature. Is there a dose-dependent effect of adipose-selective loss of HK2 on glucose homeostasis that is not apparent in global heterozygous mice?

2. Pg. 4-5, "…. expression of the glucose transporter GLUT4 was not affected in vWAT within 4 weeks of HFD (Figure 1A and Figure S1L)." Although the data in Figure 1A supports this conclusion in vWAT, in Figures S1L it appears there is significant down regulation of GLUT4 at 1 and 2 weeks on HFD in vWAT. Although there is some recovery at week 4, Glut4 levels still appear lower than at week 0. These and other blots throughout the manuscript should be quantified. What is the level of GLUT4 expression in BAT? If GLUT4 is diminished in this time frame, this may require significant revision of the authors' conclusions and discussion.

3. Figure 1C; G6P, F6P, and F1,6P are not significantly reduced in adipose tissue with HFD which argues against HK2 as the pivotal enzymatic step controlling the rate of glucose flux into adipose in this condition. Measurement of static metabolite levels are likely not sufficient to draw any firm conclusions on this and the authors should consider flux analysis using stable isotopes which may be much more informative.

4. The authors note that multiple species of cavefish are hyperglycemic and identify a variant in Molino cavefish in the HK2 gene (Figure 2) with reduced activity in vitro. As the authors note, independently evolved Cavefish have repeatedly developed distinct mutations in genes regulating diverse aspects of metabolism and fuel homeostasis including mutations in the insulin receptor and MC4R to survive their nutrient depleted environment. The authors suggest that the HK2 mutation may mediate the hyperglycemia in the Molino sub-species. While the hypothesis is interesting and testable, no experiments are performed in this manuscript that tests the role of HK2 mutant in the glycemic phenotype. As presented, the presence of the HK2 mutation in hyperglycemic fish is entirely correlative and does not provide strong support for the conclusion that HK2 plays a causal role in hyperglycemia.

5. Pg. 9, line 201: "However, adipose tissue accounts for only <5% of glucose disposal (Jackson et al., 1986; Kowalski and 203 Bruce, 2014)." These estimates do not consider the role of BAT in glucose disposal which is likely substantial in mice as suggested by the author's own data (Figure 4A). How much of the phenotype is driven by reduced BAT glucose uptake? BAT glucose uptake may be less relevant in larger animals including humans which may diminish the importance of these findings for human insulin resistance.

6. To support the hypothesis that the incomplete reductions in HK2 activity that are observed with high-fat diet in mice or obesity and diabetes in humans are important, the authors should determine with adipose HK2 heterozygosity is sufficient to impair glucose homeostasis.

7. The data regarding the HK variant in Molino cavefish is intriguing, but correlative and preliminary. It doesn't substantially help the key arguments. The authors should consider crossing Molino fish with surface fish to determine whether the hyperglycemic phenotype segregates with the HK2 mutation.

8. Figure 1C, 3E. The authors should show heat maps for metabolite levels for all animals rather than an aggregate fold change. It is difficult to interpret these results without some idea of the variance in the measures across animals. Were statistics for metabolites corrected for multiple comparisons?

9. Pg 5, line 112: "Importantly, hexokinase activity inversely correlated with insulin resistance in obese non-diabetic patients (Figure 1F)." The data shows that mean HK activity is ~ 30% lower in people with high HOMA-IR compared to low HOMA-IR. Correlation was not tested. However, given the substantial overlap in HK activity between groups, this is worth examining. Is the variance in HK activity a major 'contributor' to the variance in HOMA-IR by regression?

10. Figure 3E. Is this in fed or fasted mice? Does feeding status make a difference in these metabolite levels? The significant difference in pyruvate, but not upstream metabolites suggest important regulation in proximity to pyruvate production or catabolism, and not necessarily at the HK2 step.

11. What was the insulin infusion rate during the hyperinsulinemic-euglycemic clamps?

12. Adipose HK2 KO increases endogenous glucose production and appears to increase hepatic ChREBP activity along with ChREBP targets. Is this increase in ChREBP activity due to increased liver glucose uptake as a result of impaired WAT/BAT glucose uptake – i.e. shunting of glucose to the liver? Activation of ChREBP in the liver has been shown to regulate G6Pc expression and drive glucose production (PMID: 27669460). Could this be an alternative explanation for the increased glucose production in adipose HK2 KO mice?

[Editors’ note: further revisions were suggested prior to acceptance, as described below.]

Thank you for resubmitting your work entitled "Diet-induced loss of adipose Hexokinase 2 triggers hyperglycemia" for further consideration by *eLife*. Your revised article has been evaluated by Carlos Isales (Senior Editor) and a Reviewing Editor.

The manuscript has been improved but there are some remaining issues that need to be addressed, as outlined below:

There are a number of concerns raised by the reviewers that will only require text or format changes that you may wish to consider. The editors feel you should carefully consider these and make changes to the text that you deem appropriate in revising the interpretations and provide more caution in interpretations. See reviewers' comments.

One issue that does seem to be critical is to directly address the issue that your HK2 KO and even the Het KO reduce HK2 more than obesity in humans (nearly 100% and about 70%, respectively vs only about 30% in obesity). Further, the improvement in glucose tolerance is very small in the hetKO, which brings into question whether the smaller decrease in HK2 in obesity does actually have a significant contribution to glucose tolerance. No ITT is provided for the hetKO either. Again, if you do not wish to provide further data on this issue, the text should clearly note this result in the Results section and discuss this problem in the Discussion, and include caution on the significance of the decrease in HK2 in obesity in contributing to insulin resistance in humans.

*Reviewer #1 (Recommendations for the authors):*

This is an excellent study and the authors have properly responded to most of the comments of the reviewers. The study extends previous reports that genetic disruptions of glucose metabolism in adipose (WAT and BAT) affect whole-body glucose homeostasis. Prior studies include KO of GLUT4, RAB10, and ChREBP. Of note, KO of RAB10 in brown fat (UCP1 CRE) also induces whole-body glucose dysregulation. The RAB10 KO studies (PMID: 34303022, PMID: 27207531) should be included in the discussion because RAB10 KO causes an approximate 50% reduction in insulin-stimulated glucose uptake yet still has a large whole-body metabolic impact, in line with the argument the authors make about obesity/HFD causing a partial reduction in HK2 that has large whole body effect.

Some comments for the authors to consider:

1. There are a couple of instances in which the description of the data does not completely jive with the data presented. For example, on page 3, they state "Consistent with reduced HK2 expression, hexokinase activity, and downstream glycolytic metabolites were decreased in.…". However, the data in Figure 1d do not support a change in glycolic metabolites. Only 2PG is reduced. Strikingly, there is no reduction in G6P, the product of HK2, and perhaps the metabolite most likely to be changed. Just looking at the data, one would conclude that HFD has little effect on glycolic metabolites. I realize there are potential issues with steady-state versus flux measurements but it doesn't seem appropriate to conclude a change in glycolytic intermediates because that is what is expected if the data do not support the conclusion. If the method is not appropriate to measure a difference, then the steady-state data should be removed from the manuscript or flux performed.

Similarly, I am not sure how to interpret the data in Figure 1G, H, and I. The data in H show a statistically significant difference between "high" and "low" HOMO-IR groups in HK activity, although the effect size is very small. The text refers to severe and mild insulin resistance (HOMO-IR) groupings. What is the homo-IR cutoff for this distinction? Is this accepted or is it based on the structure of this data (e.g., quartiles)? What do the data for HK activity look like for the obese diabetic group (from panel G)? Are those all low HK activity?

The Pearson's correlation in panel I, although trending to a negative correlation, is not significant, which I believe is what one would expect if HK activity was not the only contributor to HOMO-IR. However, the authors suggest the failure to achieve statistical significance is because the measurement (HK activity rather than HK2 amount) is insufficient, which then begs the question of the validity of the measurement for the analyses in panel H.

I do not mean to be nit-picking here but I believe the major impact of the paper rests on linking a reduction in HK2 to obesity/insulin resistance since, as noted above, it is known that disruptions in glucose metabolism of adipocytes induce whole-body glucose intolerance. Re-describing and or re-analyzing the human data seems in order.

2. I agree with the authors that the Molino fish data are "cool" but they also disrupt the flow of the work (and as noted in the previous review they are somewhat "superficial" from the metabolism perspective). Might I suggest they move these data to the end of the result section, where they will not disrupt the flow and be appreciated as further support for the hypothesis linking HK2 to hyperglycemia?

3. I think the 3T3-L1 experiments are of limited value and could be removed without impacting the conclusions of the work. I might be missing the point of these data but I think all they show is that HK2 has a role in in vitro differentiated adipocytes. I am not sure how that impacts the conclusions of the study.

*Reviewer #2 (Recommendations for the authors):*

The response to previous reviews (by other reviewers) seems highly appropriate to this reviewer. Incurring in the analysis of heterozygotic adipocyte HK2 mice addressed major points brought in during that first review. There is an important point that however requires consideration.

This reviewer would only like to raise the following points that could be attended to with appropriate discussion in the text:

1. The findings presented do not support that there is a defect in glucose transport in adipocytes. There is a reduction in glucose deposition in adipose tissue during a hyperinsulinemic clamp of adipose-specific HK2 KO mice (Figure 5) and there is lower insulin-stimulated 2-deoxyglucose deposition in insulin-stimulated 3T3-L1 adipocytes depleted of HK2. However, there is no change in glucose-6-phosphate levels in adipose tissue of 4-week HFD-fed mice (Figure 1 D) and there is also no difference in the basal 2-deoxyglucose deposition in the HK2-depleted 3T3-L1 adipocytes (Figure 3E). Since 2-deoxyglucose deposition depends on hexokinase activity, it is not clear why the reduction was only seen in response to insulin. These findings deserve some comment, and the end of the 2nd paragraph of page 13 as well as the 1st paragraph of the Discussion could also be accordingly amended.

2. The above findings also mean that the model in figure 8 must be modified (it currently shows less glucose transport into adipocytes). The model should also include the other two possibilities for crosstalk with the liver mentioned on page 12 (3rd paragraph of Discussion).

3. Figure 1: Please indicate in the legend to 1A that mRNA levels are being quantified.

4. Please indicate in the y-axis of Figure 7A that the results refer to HK2 (the figures identify all other genes in the other panels).

5. Supplemental Figure S2D: The levels of GLUT4 in the immunoblot do not jive with the quantifications of the 5 n's. It would be ideal if a more representative immunoblot were shown.

[Editors’ note: further revisions were suggested prior to acceptance, as described below.]

Thank you for resubmitting your work entitled "Diet-induced loss of adipose Hexokinase 2 triggers hyperglycemia" for further consideration by *eLife*. Your revised article has been evaluated by Carlos Isales (Senior Editor) and a Reviewing Editor.

The manuscript has been improved but there is one last remaining issue that needs to be addressed, as outlined below:

The authors appropriately clarified in the Results section and in the Discussion section that the experimental perturbations in reducing HK2 were greater than that caused by obesity. They also included caution in the interpretation and now correctly used the word "may" to describe the effect of HK2 in obesity on hyperglycemia. However, the title remains a definitive statement, which does not seem appropriate in the absence of further definitive data, which is not provided. Please consider providing a more general title.

---

## [Author Response]

[Editors’ note: the authors resubmitted a revised version of the paper for consideration. What follows is the authors’ response to the first round of review.]

Reviewer #1 (Recommendations for the authors):This is an interesting manuscript which reports effects of adipose HK2 KO in an attempt to determine the role of hexokinase in adipocytes as it relates to whole body metabolism. The data indicate that VAT HK2 is strongly downregulated in obesity/HFD, while SAT shows some decrease as well, although less of an effect than seen in VAT. The adipose tissue selective KO causes marked changes in systemic glucose tolerance. The authors conclude that this ability of adipose HK2 to modify glucose tolerance is due to its requirement for the esterification of FA that is released in lipolysis. Therefore, in HK2KO adipocytes, more FA is released into the circulation and is hypothetically able to induce gluconeogenesis by known mechanisms. The following two most important concerns are noted:1. It is not surprising that HK2 KO in adipocytes causes glucose intolerance, as previous work has clearly demonstrated that inhibiting glucose metabolism in adipocytes has this consequence. In this regard, adipocyte GLUT4KO shows a similar phenotype and this has been studied in great detail in the past. Similar to GLUT4, the authors show here that HK2 is also decreased in obesity, and therefore could contribute to decreasing adipocyte glucose metabolism and the effect on glucose tolerance. However, a full KO of HK2 does not mimic the physiological system in obesity since HK2 is diminished but not totally inhibited in the obese state in humans and in the SAT of mice, while the KO completely eliminates the enzyme. Since glucose transport is considered to be rate limiting for glucose uptake, the argument hinges on whether the level of decrease in HK2 in obesity is actually contributing much to a decrease in glucose metabolism compared to the contribution of the decreased GLUT4. A much more informative experiment would be to study het mice where HK2 might be decreased to a more physiological level that is actually seen in obesity rather than the full KO where the result is difficult to interpret.

We addressed this concern by characterizing adipose-specific heterozygous HK2 knockout mice. The partial loss of adipose HK2 causes glucose intolerance (Figure S5).

2. The text goes beyond the data in attributing correlation to causation. Examples are Figure 5 and Figure 6 legends that ascribe defined mechanisms to what are essentially correlations that are observed. It is not shown definitively in the data of Figure 5, for example, that de novo lipogenesis deficiency causes increased hepatic gluconeogenesis. As a matter of fact, hepatic gluconeogenesis is not measured in this Figure. And the NEFA data are correlative with the decreased adipose tissue de novo lipogenesis. Similarly, Figure 6 doesn't show mechanism, only correlation.

Relying on known mechanisms (PMID: 29069585, 25662011, and 27238637), we concluded that increased NEFA release and decreased adipose lipogenesis contribute to increased hepatic gluconeogenesis in adipose-specific *Hk2* knockout (AdHk2KO) mice. We agree with this reviewer that we do not provide direct evidence that goes beyond correlation in the current study. Thus, we softened our conclusion for these figures (now Figure 6). In the Discussion, we discuss how loss of adipose HK2 might be linked to increased hepatic glucose production and thus glucose intolerance in AdHk2KO mice (Page 12).

Reviewer #2 (Recommendations for the authors):It is well established that feeding mice with western diets causes insulin resistance in liver, adipose tissue and muscle. There are many studies that propose various mechanisms for this effect. This paper describes a completely new and exciting mechanism involving changes in the expression of hexokinase 2, an important glycolytic enzyme, in adipose tissue. The authors provide sufficient evidence to show that the expression of hexokinase 2 in adipose tissue is able to regulate whole body metabolism. However, there are some issues that the authors could address prior to further consideration of the study. For example, further information that might address the mechanism for the fall in hexokinase 2 would be valuable and an explanation for the disparity between some of the measures of insulin sensitivity are also necessary.

Please find our response to these concerns directly below.

This is a terrific study that brings new light on this heavily studied problem of metabolic dysfunction in the high fat fed mouse. The data is for the most part of very high quality and the manuscript is well written and researched.

We thank the reviewer for finding our study ‘terrific’ and of ‘very high quality.’

1. The biggest issue for me with this paper is the huge discrepancy between the 'clamp' data and the other measures of insulin action. I have personally never seen such a massive defect in insulin action by clamp as this. It essentially shows that there is zero response to insulin whatsoever in these mice. Yet all other tests (GTT, ITT etc) show a relatively mild phenotype. The tissue specific glucose uptake is also not in step with this defect in that the liver defect is mild, there is no apparent defect in muscle and the only real defect is in adipose tissue, which theoretically accounts for a small proportion of whole body glucose utilisation. The other issue is that in absolute terms 2DG uptake in muscle is on par with WAT which seems worthy of questioning. I was unable to find the insulin levels generated during the clamp so I am not sure if that is a problem. I strongly urge the authors to go back over these data to see if they can spot a calculation error.

We went back over the data and checked calculations. We did not identify an error. As one can see in the glucose infusion data (Figure 5F and its source data), we did not need to infuse glucose in 4 out of 6 KO mice to maintain euglycemia (150 mg/dL, Figure 5E and its source data). Since reduced glucose infusion was already obvious in the raw data, it could not be the case that the defect in insulin action is due to a calculation error.

Finally, it is important to note that the conclusion drawn from our clamp data is in line with other tests. The clamp data support our conclusion that adipose-specific HK2 loss causes insulin insensitivity and glucose intolerance.

Please note that 2DG uptake is normalized to the mass of the analyzed tissue (approx.0.05-0.1 g), not to the mass of the whole tissue. Since muscle mass (lean mass, 70-80% of body weight) is much larger than WAT (1-2% of bodyweight) and BAT (0.5% of bodyweight) mass, our data still show that the majority of 2DG (glucose) is disposed in muscle. This is mentioned in the legend of figure 5A.

2. Given that the authors have deep proteomics data from these mice it would be interesting to run a co-regulation analysis (e.g. WGCNA) to determine if there are other proteins in the data that behave like HK2, as this might provide some mechanistic hints.

As requested, we performed a co-regulation analysis in the AHA-proteome and RNAseq data to identify proteins whose expression correlated with HK2. We identified 155 proteins whose AHA incorporation was downregulated and correlated with the AHA incorporation of HK2 in WAT of HFD-fed mice. Next, we performed a pathway enrichment analysis to identify pathways regulated like HK2. The top five pathways enriched among the 155 proteins are ribosomal proteins, the electron transport chain (ETC), oxidative phosphorylation (OXPHOS), fatty acid (FA) synthesis, and glycolysis (Figure S9A). Interestingly, similar to HK2, the AHA incorporation into ribosomal proteins and proteins in ETC and OXPHOS was reduced without affecting the level of their encoding RNAs (Figure S9B), suggesting that HK2 might be regulated in a similar way as ribosomal proteins and proteins in ETC and OXPHOS. How these proteins including HK2 are post-transcriptionally regulated is for future investigation. This analysis (Figure S9) is now included in the manuscript in the last section of the Results.

3. The mechanism for the loss of insulin regulation of lipolysis in the HK Mice is interesting. I was not particularly convinced by the glycerol data as measures of plasma glycerol are not particularly informative as it probably has a very high flux. So as a minimum I think this part needs expansion.

We agree with the reviewer. Since the mechanism of how loss of HK2 causes increased glycerol release is beyond the scope of this study and is not part of our main conclusions, we removed the glycerol data from the manuscript.

4. In your AHA experiments which are very nice it would be valuable if you could show some 'control' proteins that do not behave like HK2.

We had CALX as a control protein in the submitted manuscript. It is still included, now in Figure 7B. Furthermore, in response to this request, we now include all mass spec quantified proteins in the AHA experiment together with RNA seq data as Supplemental Table S3. These data can be informative for the scientific community to further study post-transcriptionally controlled proteins in adipose tissue of HFD-fed mice.

5. It would be fascinating to know if deletion of HK2 has affected GLUT4 translocation? I know this might be a bit of a stretch for the authors but it might be easily achievable in your 3T3-L1 cells if you could express t

We respectfully agree with the review that this is a ‘stretch’ for the current study.

Reviewer #3 (Recommendations for the authors):In this manuscript, Shimobayashi et al. investigate the molecular pathogenesis of high-fat diet induced insulin resistance with a focus on adipose tissue. The authors show that short-term (4 weeks) high-fat diet is sufficient to cause glucose intolerance in male mice. Through a proteomic screen, they identify downregulation of HK2 in visceral fat as one of many changes in the adipose proteome in this timeframe. The authors observe that HK2 expression and HK activity is reduced in adipose tissue in human obesity. The authors propose that down-regulation of adipose HK2 is an early and critical step in the pathogenesis of insulin resistance. The authors also observe that the hyperglycemic Molino Cavefish sub-species carries an inactivating mutation in HK2 and propose that this contributes to hyperglycemia in this model organism. The authors generate adipose specific HK2 knockout mice and show that this is sufficient to cause peripheral insulin resistance with reduced glucose uptake in WAT and BAT as well as hepatic insulin resistance with increased glucose production. This is a well-written manuscript conducted with sophistication and technical expertise. The comprehensive phenotyping of the adipose HK2 KO mice is a strength and the phenotypes are clear and robust. However, as detailed below, previously published data not discussed in this manuscript indicates that substantial reductions in HK2 activity in vivo are insufficient to cause systemic insulin resistance and this undermines the author's conclusion that downregulation of HK2 is the critical step in high-fat diet induced changes in adipose function.Concerns:1. Prior literature (PMID: 10428828) indicates that global HK2 heterozygous mice demonstrated a 50% reduction in tissue HK2 activity (in both adipose and muscle) but have normal glucose homeostasis including normal glucose tolerance and insulin action. Although the authors show that complete ablation of adipose HK2 can impair systemic glucose metabolism, the previously published work calls in to question the physiological importance intermediate reductions in adipose HK2 activity that are commonly observed with obesity or dietary interventions. This is particularly true for humans where mean adipose HK activity is reduced by ~ 50% in people with obesity with substantial variance and overlap with those who are lean. The current work must be discussed with respect to this prior literature. Is there a dose-dependent effect of adipose-selective loss of HK2 on glucose homeostasis that is not apparent in global heterozygous mice?

We thank the reviewer for pointing this out. We addressed this concern by characterizing adipose-specific heterozygous HK2 knockout mice. The heterozygous KO mice are glucose intolerant, compared to controls. This information is now added to the manuscript (Figure S5).

Indeed, a previous study demonstrated that global heterozygous *Hk2* knockout mice have normal, even better, glucose tolerance compared to controls [PMID: 10428828]. How can we explain this apparent discrepancy? We note that adipose-specific GLUT4 knockout mice display glucose intolerance [PMID: 11217863], but global GLUT4 knockout mice are glucose tolerant due to a compensatory increase in glucose uptake in liver [PMID: 7675081, 15793230]. We observed increased systemic glycolysis in AdHk2KO mice (see Figure S4N). Global ablation of HK2 mice may provoke an even stronger compensatory response to maintain systemic glucose homeostasis. This may also explain the observation that global heterozygous *Hk2* knockout mice display better glucose handling than wild-type controls [PMID: 10428828]. This text has now been added to the Discussion in the revised manuscript.

2. Pg. 4-5, "…. expression of the glucose transporter GLUT4 was not affected in vWAT within 4 weeks of HFD (Figure 1A and Figure S1L)." Although the data in Figure 1A supports this conclusion in vWAT, in Figures S1L it appears there is significant down regulation of GLUT4 at 1 and 2 weeks on HFD in vWAT. Although there is some recovery at week 4, Glut4 levels still appear lower than at week 0. These and other blots throughout the manuscript should be quantified. What is the level of GLUT4 expression in BAT? If GLUT4 is diminished in this time frame, this may require significant revision of the authors' conclusions and discussion.

As requested, we have quantified immunoblots including those indicated above. GLUT4 expression in vWAT was slightly reduced at 4 weeks of HFD feeding. Also, we have included immunoblots of BAT as requested. GLUT4 expression in BAT was slightly reduced, but significantly only at 2 weeks of HFD. In all fat depots, we observed that HK2 expression is decreased more than GLUT4 expression, supporting our claim that HK2 expression correlates with diet-insulin insensitivity.

3. Figure 1C; G6P, F6P, and F1,6P are not significantly reduced in adipose tissue with HFD which argues against HK2 as the pivotal enzymatic step controlling the rate of glucose flux into adipose in this condition. Measurement of static metabolite levels are likely not sufficient to draw any firm conclusions on this and the authors should consider flux analysis using stable isotopes which may be much more informative.

We agree with the reviewer that measuring static levels of glycolytic and PPP metabolites is not sufficient to make a firm conclusion. However, we present reduced glycolytic and PPP metabolites as additional supporting evidence for diet-induced HK2 downregulation in adipose tissue of HFD-fed mice, in addition to downregulation of HK2 expression itself (Figures 1A, 1B, 1E, 1F) and tissue hexokinase activity (Figure S1F).

4. The authors note that multiple species of cavefish are hyperglycemic and identify a variant in Molino cavefish in the HK2 gene (Figure 2) with reduced activity in vitro. As the authors note, independently evolved Cavefish have repeatedly developed distinct mutations in genes regulating diverse aspects of metabolism and fuel homeostasis including mutations in the insulin receptor and MC4R to survive their nutrient depleted environment. The authors suggest that the HK2 mutation may mediate the hyperglycemia in the Molino sub-species. While the hypothesis is interesting and testable, no experiments are performed in this manuscript that tests the role of HK2 mutant in the glycemic phenotype. As presented, the presence of the HK2 mutation in hyperglycemic fish is entirely correlative and does not provide strong support for the conclusion that HK2 plays a causal role in hyperglycemia.

We agree with the reviewer. We do not claim that the HK2 variant is the driver for hyperglycemia in Molino cavefish. We are currently breeding surface and Molino cavefish to perform a quantitative trait (QTL) analysis. However, Mexican tetra become sexually mature and can be mated after 8 months but ideally at more than 1 year old, and Molino cavefish are especially difficult to produce (PMID: 30638199). Taking these challenges into account, we need a minimum of 2-3 years of investigations to complete the QTL analysis. In the current manuscript, we only conclude that the HK2 variant (R42H) found in Molino is a loss-of-function mutation based on our enzyme assay (Figures 2F-2G) and add that it is supporting orthogonal evidence that HK2 loss affects glucose homeostasis. Also, it is a very cool result.

5. Pg. 9, line 201: "However, adipose tissue accounts for only <5% of glucose disposal (Jackson et al., 1986; Kowalski and 203 Bruce, 2014)." These estimates do not consider the role of BAT in glucose disposal which is likely substantial in mice as suggested by the author's own data (Figure 4A).

Note that the glucose uptake rate in Figure 4A (now in Figure 5A) is normalized to the tissue mass (mg tissue) tested in the assay. When we calculate glucose uptake rate per depot (based on glucose uptake and tissue weight) in control mice, the glucose uptake rate is 7.5 ug/min in WAT and 13.6 ug/min in BAT. Since the number of WAT depots is more than that of BAT depots in humans and even in mice, we believe that the contribution of BAT in glucose disposal is almost identical to that of WAT. We have not added this information to the manuscript but can do so if the reviewer insists.

How much of the phenotype is driven by reduced BAT glucose uptake?

This is an important question, and we are currently generating *Ucp1-Cre, Hk2^fl/fl^* mice to answer this question. We believe that characterization of BAT-specific HK2 knockout mice is beyond the scope of the current study.

BAT glucose uptake may be less relevant in larger animals including humans which may diminish the importance of these findings for human insulin resistance.

Please see the above.

6. To support the hypothesis that the incomplete reductions in HK2 activity that are observed with high-fat diet in mice or obesity and diabetes in humans are important, the authors should determine with adipose HK2 heterozygosity is sufficient to impair glucose homeostasis.

As requested, we have examined heterozygous knockout mice and these new experiments have been added to the manuscript. Please see our reply point 1 above.

7. The data regarding the HK variant in Molino cavefish is intriguing, but correlative and preliminary. It doesn't substantially help the key arguments. The authors should consider crossing Molino fish with surface fish to determine whether the hyperglycemic phenotype segregates with the HK2 mutation.

We believe that identification of the loss-of-function *Hk2* variant in Molino cavefish is an important finding although it is indeed correlative at this stage. Also, please see our reply to the point 4 above.

8. Figure 1C, 3E. The authors should show heat maps for metabolite levels for all animals rather than an aggregate fold change. It is difficult to interpret these results without some idea of the variance in the measures across animals. Were statistics for metabolites corrected for multiple comparisons?

As requested, we now changed the heat maps to bar graphs. The conclusion is the same as before. We performed Student’s t-test for each metabolite since we normalized each metabolite in KO to that in control and do not compare different metabolites.

9. Pg 5, line 112: "Importantly, hexokinase activity inversely correlated with insulin resistance in obese non-diabetic patients (Figure 1F)." The data shows that mean HK activity is ~ 30% lower in people with high HOMA-IR compared to low HOMA-IR. Correlation was not tested. However, given the substantial overlap in HK activity between groups, this is worth examining. Is the variance in HK activity a major 'contributor' to the variance in HOMA-IR by regression?

As requested, we calculated Pearson’s R between HOMA-IR and HK activity in obese non-diabetic patients. We observed only a trend toward negative correlation. This is now added to the manuscript (see Figure 1I). However, we also mention in the text that, “the lack of statistical significance is possibly due to the fact that, in the absence of an antibody that recognizes human adipose HK2, we measured total hexokinase activity rather than specifically HK2 expression or activity.” The fact that we measured total HK activity most likely masked some of the effect of loss of specifically HK2.

10. Figure 3E. Is this in fed or fasted mice? Does feeding status make a difference in these metabolite levels? The significant difference in pyruvate, but not upstream metabolites suggest important regulation in proximity to pyruvate production or catabolism, and not necessarily at the HK2 step.

The data in Figure 3E (now Figure 4B) are from ad libitum-fed mice. In this study, the metabolomics data are provided to support reduced HK2 activity (Figure 3B), glucose uptake (Figures 3E and 5A), and glycolysis (Figure 3D) in adipose tissue/adipocytes of the HK2 KO/KD. We agree that the metabolomics alone cannot firmly allow the conclusion that glycolysis is downregulated in adipose tissue of AdHk2KO mice. However, all the data taken together, strongly support the conclusion that loss of adipose HK2 causes reduced glucose influx in adipose tissue/adipocytes.

11. What was the insulin infusion rate during the hyperinsulinemic-euglycemic clamps?

The insulin infusion rate was 0.09 mU/min. This information is now included in the method section (Page 24).

12. Adipose HK2 KO increases endogenous glucose production and appears to increase hepatic ChREBP activity along with ChREBP targets. Is this increase in ChREBP activity due to increased liver glucose uptake as a result of impaired WAT/BAT glucose uptake – i.e. shunting of glucose to the liver? Activation of ChREBP in the liver has been shown to regulate G6Pc expression and drive glucose production (PMID: 27669460). Could this be an alternative explanation for the increased glucose production in adipose HK2 KO mice?

We thank the reviewer for bringing this to our attention. We agree that glucose shunting to the liver due to impaired WAT/BAT glucose uptake in the KO mice can be alternative mechanism for the increased glucose production in the KO mice. This mechanism is now included in the Discussion (Page 12).

[Editors’ note: what follows is the authors’ response to the second round of review.]

One issue that does seem to be critical is to directly address the issue that your HK2 KO and even the Het KO reduce HK2 more than obesity in humans (nearly 100% and about 70%, respectively vs only about 30% in obesity). Further, the improvement in glucose tolerance is very small in the hetKO, which brings into question whether the smaller decrease in HK2 in obesity does actually have a significant contribution to glucose tolerance. No ITT is provided for the hetKO either. Again, if you do not wish to provide further data on this issue, the text should clearly note this result in the Results section and discuss this problem in the Discussion, and include caution on the significance of the decrease in HK2 in obesity in contributing to insulin resistance in humans.

In the Results section, we now indicate the % reduction in HK2 expression in HFD-fed mice, AdHk2KO mice and the % reduction in HK activity in obese/diabetic patients.

Now it reads;

Page 3, “Quantification by immunoblotting revealed an approximately 60%, 90% and 50% reduction in HK2 expression in vWAT, subcutaneous WAT (sWAT), and brown adipose tissue (BAT), respectively, of 4-week HFD mice (Figures 1B-C).”

Page 4, “Similar to our findings in mice, omental WAT (human vWAT) biopsies from obese non-diabetic and obese diabetic patients displayed a ~30% reduction in hexokinase activity (Figure 1G and Supplementary file 2). ”Page 6,

“In AdHk2KO mice, HK2 expression was decreased ~75% in vWAT, ~70% in sWAT, and ~65% in BAT but unchanged in skeletal muscle (Figure 4A and Figure 4 —figure supplements 1B-D).“

-> As requested, we now include a paragraph in the Discussion section dealing with the issue that our HK2 KO does not completely phenocopy adipose tissue of obese mice and patients.

It reads;

Page 13, “We note that *Hk2* knockout did not completely phenocopy to the effect of obesity in mice and humans. While HK2 was decreased approximately 75% in vWAT of AdHk2KO mice (Figure 4A), HK2 expression in HFD-fed mice and HK activity in oWAT from obese patients were decreased only ~60% and ~30%, respectively (Figures 1B and 1G). Is the reduction in HK2 in obese mice and patients sufficient to contribute to systemic insulin insensitivity and glucose intolerance? A previous study showed that adipose-specific *Rab10* knockout mice display reduced GLUT4 translocation to the plasma membrane and thus a ~50% reduction in insulin-stimulated glucose disposal, in isolated adipocytes (Vazirani et al., 2016). Importantly, adipose-specific *Rab10* knockout mice failed to suppress hepatic glucose production and were thus insulin insensitive, indicating that a limited disruption in glucose metabolism in adipose tissue can significantly impact systemic insulin sensitivity and glucose homeostasis. Thus, the partial loss of HK2 observed in obese mice and patients may be sufficient to impact systemic insulin sensitivity and thereby glucose homeostasis.”

Reviewer #1 (Recommendations for the authors):This is an excellent study and the authors have properly responded to most of the comments of the reviewers. The study extends previous reports that genetic disruptions of glucose metabolism in adipose (WAT and BAT) affect whole-body glucose homeostasis. Prior studies include KO of GLUT4, RAB10, and ChREBP. Of note, KO of RAB10 in brown fat (UCP1 CRE) also induces whole-body glucose dysregulation. The RAB10 KO studies (PMID: 34303022, PMID: 27207531) should be included in the discussion because RAB10 KO causes an approximate 50% reduction in insulin-stimulated glucose uptake yet still has a large whole-body metabolic impact, in line with the argument the authors make about obesity/HFD causing a partial reduction in HK2 that has large whole body effect.

We thank the reviewer for this suggestion. We now cite and discuss the study on adipose-specific *Rab10* knockout mice (PMID: 27207531) suggesting that a limited reduction in insulin-stimulated glucose disposal in adipose tissue impact systemic insulin sensitivity and glucose homeostasis (see above). We did not include the other paper (PMID: 34303022), because this other study focuses on characterization of brown adipose tissue (BAT)-specific RAB10 knockout mice and we do not characterize a BAT-specific HK2 knockout mice in the current study.

Some comments for the authors to consider:1. There are a couple of instances in which the description of the data does not completely jive with the data presented. For example, on page 3, they state "Consistent with reduced HK2 expression, hexokinase activity, and downstream glycolytic metabolites were decreased in.…". However, the data in Figure 1d do not support a change in glycolic metabolites. Only 2PG is reduced. Strikingly, there is no reduction in G6P, the product of HK2, and perhaps the metabolite most likely to be changed. Just looking at the data, one would conclude that HFD has little effect on glycolic metabolites. I realize there are potential issues with steady-state versus flux measurements but it doesn't seem appropriate to conclude a change in glycolytic intermediates because that is what is expected if the data do not support the conclusion. If the method is not appropriate to measure a difference, then the steady-state data should be removed from the manuscript or flux performed.

As requested, we removed the data of Figure 1D and modified the text accordingly.

Similarly, I am not sure how to interpret the data in Figure 1G, H, and I. The data in H show a statistically significant difference between "high" and "low" HOMO-IR groups in HK activity, although the effect size is very small. The text refers to severe and mild insulin resistance (HOMO-IR) groupings. What is the homo-IR cutoff for this distinction? Is this accepted or is it based on the structure of this data (e.g., quartiles)? What do the data for HK activity look like for the obese diabetic group (from panel G)? Are those all low HK activity?

As indicated in the legend of Figure 1H, the HOMA-IR cut-off of severe and mild insulin resistance is 2.9, which was described as a median value in diabetic patients (PMID:3899825).

We have also performed similar analyses for the obese diabetic group (see Author response image 1). Although most patients with HOMA-IR>2.9 displayed low HK2 activity, there is only one patient with HOMA-IR<2.9 in the obese and diabetes group. Since we do not have sufficient patient numbers to draw a solid conclusion in this group, we do not include this analysis in the manuscript.

**Author response image 1. sa2fig1:** 

The Pearson's correlation in panel I, although trending to a negative correlation, is not significant, which I believe is what one would expect if HK activity was not the only contributor to HOMO-IR. However, the authors suggest the failure to achieve statistical significance is because the measurement (HK activity rather than HK2 amount) is insufficient, which then begs the question of the validity of the measurement for the analyses in panel H.I do not mean to be nit-picking here but I believe the major impact of the paper rests on linking a reduction in HK2 to obesity/insulin resistance since, as noted above, it is known that disruptions in glucose metabolism of adipocytes induce whole-body glucose intolerance. Re-describing and or re-analyzing the human data seems in order.

We have modified the text describing the Figure 1I, as suggested.

Now it reads on page 4;

“Although hexokinase activity negatively correlated with insulin resistance in obese patients, this correlation was not statistically significant (Figure 1I), suggesting that loss of hexokinase activity may not be the only cause of insulin resistance in human.”

2. I agree with the authors that the Molino fish data are "cool" but they also disrupt the flow of the work (and as noted in the previous review they are somewhat "superficial" from the metabolism perspective). Might I suggest they move these data to the end of the result section, where they will not disrupt the flow and be appreciated as further support for the hypothesis linking HK2 to hyperglycemia?

The Molino data provide an important link between reduced HK2 activity and hyperglycemia, similar to adipose tissue of obese mice. Thus, we believe that describing the Molino data before generating and characterizing adipose-specific *Hk2* knockout mice is appropriate.

3. I think the 3T3-L1 experiments are of limited value and could be removed without impacting the conclusions of the work. I might be missing the point of these data but I think all they show is that HK2 has a role in in vitro differentiated adipocytes. I am not sure how that impacts the conclusions of the study.

The 3T3-L1 data are more supporting evidence that HK2 is important for glucose disposal in adipocytes. Thus, we prefer to keep the 3T3-L1 data as they are.

Reviewer #2 (Recommendations for the authors):The response to previous reviews (by other reviewers) seems highly appropriate to this reviewer. Incurring in the analysis of heterozygotic adipocyte HK2 mice addressed major points brought in during that first review. There is an important point that however requires consideration.This reviewer would only like to raise the following points that could be attended to with appropriate discussion in the text:1. The findings presented do not support that there is a defect in glucose transport in adipocytes. There is a reduction in glucose deposition in adipose tissue during a hyperinsulinemic clamp of adipose-specific HK2 KO mice (Figure 5) and there is lower insulin-stimulated 2-deoxyglucose deposition in insulin-stimulated 3T3-L1 adipocytes depleted of HK2. However, there is no change in glucose-6-phosphate levels in adipose tissue of 4-week HFD-fed mice (Figure 1 D) and there is also no difference in the basal 2-deoxyglucose deposition in the HK2-depleted 3T3-L1 adipocytes (Figure 3E). Since 2-deoxyglucose deposition depends on hexokinase activity, it is not clear why the reduction was only seen in response to insulin. These findings deserve some comment, and the end of the 2nd paragraph of page 13 as well as the 1st paragraph of the Discussion could also be accordingly amended.

We do not argue that there is a defect in glucose ‘transport’ in HK2-depleted adipocytes. Indeed, we conclude, “loss of HK2 decreases glucose disposal in insulin-stimulated adipocytes” (Line 5 on page 6).

Regarding the observation that there is no difference in basal 2-deoxyglucose “deposition” in HK2 knockdown adipocytes compared to controls, it is well established that insulin promotes HK2 activity by promoting translocation of HK2 to mitochondria where HK2 uses ATP exiting mitochondria to phosphorylate glucose (PMID: 18064042, 11390360). Thus, in the absence of insulin, HK2 is inactive in control adipocytes, and we do not expect to see the deference in basal 2-deoxyglucose disposal in HK2 knockdown adipocytes. To clarify this point, we modified the text as follows on pages 6:

“Furthermore, although basal glucose accumulation did not differ, insulin-stimulated glucose accumulation was 50% lower in HK2-knockdown adipocytes (Figure 3E), despite normal insulin signaling (Figure 3A). The observation that HK2-knockdown has not effect on basal glucose accumulation is due to the fact that HK2 is inactive in the absence of insulin (Gottlob et al., 2001; Miyamoto, Murphy, and Brown, 2008). Thus, loss of HK2 decreases glucose disposal in insulin-stimulated adipocytes in vitro.”

2. The above findings also mean that the model in figure 8 must be modified (it currently shows less glucose transport into adipocytes). The model should also include the other two possibilities for crosstalk with the liver mentioned on page 12 (3rd paragraph of Discussion).

Figure 8 reflects less glucose disposal, not less glucose transport. To increase clarity, we modified the figure legend. It now reads:

“B. Diet-induced loss of adipose HK2 triggers glucose intolerance via reduced glucose disposal in adipocytes (left) and de-repressed hepatic gluconeogenesis despite maintained lipogenesis (right).” We refrain from including the other two possible models (the control of hepatic glucose production by neuronal inputs or increased glucose uptake in the liver) in Figure 8 since we did not test them in this study.

3. Figure 1: Please indicate in the legend to 1A that mRNA levels are being quantified.

Figure 1A is a quantification of the proteome data. We modified the legend and it now reads;

“A. The Log_2_ fold change (FC) of Hexokinase and GLUT4 protein expression in visceral white adipose tissue (vWAT) of normal diet (ND)- and 4-week high fat diet (HFD)-fed wild-type C57BL/6JRj mice. Multiple t test, **q<0.0001. n=5 (ND) and 5 (HFD).”

4. Please indicate in the y-axis of Figure 7A that the results refer to HK2 (the figures identify all other genes in the other panels).

The y-axis of Figure 7A refers to and has been already indicated as *Hk2 mRNA.*

5. Supplemental Figure S2D: The levels of GLUT4 in the immunoblot do not jive with the quantifications of the 5 n's. It would be ideal if a more representative immunoblot were shown.

We now provide a representative blot.

[Editors’ note: what follows is the authors’ response to the third round of review.]

The manuscript has been improved but there is one last remaining issue that needs to be addressed, as outlined below:The authors appropriately clarified in the Results section and in the Discussion section that the experimental perturbations in reducing HK2 were greater than that caused by obesity. They also included caution in the interpretation and now correctly used the word "may" to describe the effect of HK2 in obesity on hyperglycemia. However, the title remains a definitive statement, which does not seem appropriate in the absence of further definitive data, which is not provided. Please consider providing a more general title.

As requested, we changed the title to “Diet-induced loss of adipose Hexokinase 2 correlates with hyperglycemia.”